# LUDVIG: Learning-free Uplifting of 2D Visual features to Gaussian Splatting scenes

## Abstract

We address the problem of extending the capabilities of vision foundation models such as DINO, SAM, and CLIP, to 3D tasks. Specifically, we introduce a novel method to uplift 2D image features into 3D Gaussian Splatting scenes. Unlike traditional approaches that rely on minimizing a reconstruction loss, our method employs a simpler and more efficient feature aggregation technique, augmented by a graph diffusion mechanism. Graph diffusion enriches features from a given model, such as CLIP, by leveraging pairwise similarities that encode 3D geometry or similarities induced by another embedding like DINOv2. Our approach achieves performance comparable to the state of the art on multiple downstream tasks while delivering significant speed-ups. Notably, we obtain competitive segmentation results using generic DINOv2 features, despite DINOv2 not being trained on millions of annotated segmentation masks like SAM. When applied to CLIP features, our method demonstrates strong performance in open-vocabulary, language-based object detection tasks, highlighting the versatility of our approach.

## 1 Introduction

The field of image understanding has recently seen remarkable progress, driven by large pretrained models such as CLIP (Radford et al., 2021), DINO (Caron et al., 2021; Oquab et al., 2024), or SAM (Kirillov et al., 2023). A key factor behind their exceptional generalization capabilities lies in the vast size of their training datasets, often composed of millions or even billions of samples.

3D scene representation has also advanced with machine learning approaches like NeRF (Mildenhall et al., 2021) and model fitting techniques such as Gaussian Splatting (Kerbl et al., 2023). These methods typically rely on a few dozen views of the scene captured from different angles. While the resulting reconstructions effectively capture both appearance and geometrical information, they are not directly applicable to semantic tasks, which has led to further developments.

The complementarity of these two families of approaches has indeed recently been exploited by numerous methods that integrate geometry and semantics by uplifting image-level features extracted by large pretrained models into 3D NeRF or Gaussian Splatting representations. This has led to a surge in methods for tasks such as language-guided object retrieval (Kerr et al., 2023; Liu et al., 2023; Zuo et al., 2024), scene editing, (Kobayashi et al., 2022; Chen et al., 2024; Fan et al., 2023), or semantic segmentation (Cen et al., 2023c; Ye et al., 2024a; Ying et al., 2024).

The main limitation of most previous approaches lies in their reliance on optimization, which requires an iterative process to learn a scene-specific 3D representation by minimizing reprojection error across all training views. While this loss function is intuitive, a faster and more straightforward method for transferring 2D generic visual features to *already trained* Gaussian splatting 3D models would be preferable, which is the purpose of this work.

In this paper, we demonstrate that a simple, learning-free process is highly effective for uplifting 2D features or semantic masks into 3D Gaussian Splatting scenes. This process, which can be viewed as an 'inverse rendering' operation, is both computationally efficient and adaptable to any feature type. We showcase its efficiency by uplifting visual features from DINOv2 (Oquab et al., 2024; Darcet et al., 2024), semantic masks from SAM (Kirillov et al., 2023) and SAM2 (Ravi et al., 2024), and language features from CLIP (Ilharco et al., 2021). Then, we show that a graph diffusion mechanism (Kondor & Lafferty, 2002; Smola & Kondor, 2003) is helpful for feature uplifting in

3D scenes. This mechanism is rooted in spectral graph theory and used in spectral clustering techniques (Belkin & Niyogi, 2001; Shi & Malik, 2000; Meila & Shi, 2000). In the context of our work, it serves multiple purposes: first, it enriches 3D features obtained from a given model such as CLIP with 3D geometry, and it may leverage rich features embeddings such as DINOv2. Second, graph diffusion transforms coarse segmentation inputs, such as scribbles, into accurate 3D segmentation masks without relying on segmentation models like SAM. When evaluated on segmentation and open-vocabulary object localization, our method achieves results comparable to state-of-the-art techniques while being significantly faster than previous approaches relying on optimization.

To summarize, our contributions are threefold: (i) we introduce a simple, learning-free uplifting approach that can be directly integrated into the rendering process, achieving state-of-the-art results when applied to SAM-generated semantic masks. (ii) we demonstrate that when using graph diffusion, uplifting DINOv2 features, yields competitive segmentation results (Section 4), despite DINOv2 not being trained for segmentation like SAM. (iii) We show that graph diffusion can also be used to enrich 3D CLIP representations, leveraging similarities computed from DINOv2 features, thereby achieving competitive performance on open-vocabulary object localization tasks.

## 2 RELATED WORK

**Learning 3D semantic scene representations with NeRF.** NeRF (Mildenhall et al., 2021) uses a multilayer perceptron to predict the volume density and radiance for any given 3D position and viewing direction. Such representation can naturally be extended to semantic features. The early works N3F (Tschernezki et al., 2022) and DFF (Kobayashi et al., 2022) distill DINO 2D (*i.e.*, image-level) features (Caron et al., 2021) in scene-specific NeRF representations. Kobayashi et al. (2022) also distill LSeg (Li et al., 2022) a CLIP-inspired language-driven model for semantic segmentation. Shortly after, LERF (Kerr et al., 2023) and 3D-OVS (Liu et al., 2023) learned 3D CLIP (Radford et al., 2021) and DINO (Caron et al., 2021) features jointly for open-vocabulary segmentation. These works were extended to other pretrained models such as latent diffusion models (Ye et al., 2023) or SAM (Kirillov et al., 2023) for semantic segmentation (Cen et al., 2023c; Ying et al., 2024).

**Learning 3D semantic scene representations with Gaussian splatting.** Subsequent work have relied on the more recent Gaussian splatting method (Kerbl et al., 2023), achieving high-quality novel-view synthesis while being orders of magnitude faster that NeRF-based models. Several tasks have been addressed such as semantic segmentation using SAM (Cen et al., 2023b; Ye et al., 2024a; Kim et al., 2024), language-driven retrieval or editing using CLIP combined with DINO (Zuo et al., 2024) or SAM (Ye et al., 2023), or scene editing using diffusion models (Chen et al., 2024; Wang et al., 2024). These works learn 3D semantic representations by minimizing a reprojection loss. As a single scene can be represented by over a million Gaussians, such optimization-based techniques have strong memory and computational limitations. To handle these, FMGS (Zuo et al., 2024) employs a multi-resolution hash embedding (MHE) of the scene for uplifting DINO and CLIP representations, Feature 3DGS (Zhou et al., 2024) learns a $1 \times 1$ convolutional upsampler of Gaussians' features distilled from LSeg and SAM's encoder and LangSplat (Qin et al., 2024) learns an autoencoder to reduce CLIP feature dimension from 512 to 3. In contrast, our approach requires no learning, which significantly speeds up the uplifting process and reduces the memory requirements.

**Leveraging 3D information to better segment in 2D.** Most prior works focusing on semantic segmentation leverage 2D models specialized for this task. The early work of Yen-Chen et al. (2022) uplifts learned 2D image inpainters by optimizing view consistency over depth and appearance. Later, subsequent works have mostly relied on uplifting either features from SAM's encoder (Zhou et al., 2024), binary SAM masks (Cen et al., 2023c;b), or SAM masks automatically generated for all objects in the image (Ye et al., 2024a; Ying et al., 2024; Kim et al., 2024). The latter approach is computationally expensive, as it requires querying SAM on a grid of points over the image. It also requires matching inconsistent mask predictions across views, with *e.g.* a temporal propagation model (Ye et al., 2024a) or a hierarchical learning approach (Kim et al., 2024), which introduces additional computational overhead. In this work, we focus on single instance segmentation and show that our uplifted features are on par with state-of-the-art approaches (Cen et al., 2023c;b; Ying et al., 2024). Standing out from prior work uplifting DINO features (Tschernezki et al., 2022; Kobayashi et al., 2022; Kerr et al., 2023; Liu et al., 2023; Ye et al., 2023; Zuo et al., 2024), we quantitatively show that DINOv2 features can be used on their own for semantic segmentation and rival SAM-based models through a simple graph diffusion process that leverages 3D geometry.

**Learning 3D CLIP features for open-vocabulary object localization.** For learning 3D CLIP features, prior works leverage vision models such as DINO or SAM. DINO is used to regularize and refine CLIP features (Kerr et al., 2023; Liu et al., 2023; Zuo et al., 2024), while SAM is employed for generating instance-level CLIP representations. These approaches suffer from high computational costs, resorting to either dimensionality reduction or efficient multi-resolution embedding representations, and usually run for a total of one to two hours for feature map generation and 3D feature optimization. In contrast, our approach bypasses the high computational cost of gradient-based optimization and, combined with graph diffusion, is an order of magnitude faster than these prior works.

## 3 UPLIFTING 2D VISUAL REPRESENTATIONS INTO 3D

In this section, we present a simple yet effective method for lifting 2D visual features into 3D using Gaussian splatting and discuss its relation with more expensive optimization-based techniques.

### 3.1 BACKGROUND ON GAUSSIAN SPLATTING

**Scene representation.** The Gaussian splatting method consists in modeling a 3D scene as a set of $n$ Gaussians densities $\mathcal{N}_i$, each defined by a mean $\mu_i$ in $\mathbb{R}^3$, a covariance $\Sigma_i$ in $\mathbb{R}^{3\times3}$, an opacity $\sigma_i$ in $(0, 1)$, and a color function $c_i(d)$ that depends on the camera pose $d$.

A 2D frame at a given view is an image rendered by projecting the 3D Gaussians onto a 2D plane, parametrized by the camera pose $d$. This projection accounts for the opacity of the Gaussians and the order in which rays associated with each pixel pass through the densities. More precisely, a pixel $p$ for a view $d$ is associated to an ordered set $\mathcal{S}_{d,p}$ of Gaussians and its value is obtained by their weighted contributions:

$$\hat{I}_d(p) = \sum_{i \in \mathcal{S}_{d,p}} c_i(d) w_i(d, p). \tag{1}$$

The above weights are obtained by $\alpha$-blending, i.e. $w_i(d, p) = \alpha_i(d, p) \prod_{j \in \mathcal{S}_{d,p}, j < i} (1 - \alpha_j(d, p))$, where the Gaussian contributions $\alpha_i(d, p)$ are computed by multiplying the opacity $\sigma_i$ by the Gaussian density $\mathcal{N}_i$ projected onto the 2D plane at pixel position $p$.

**Scene optimization.** Let $I_1, \ldots, I_m$ be a set of 2D frames from a 3D scene and $d_1, \ldots, d_m$ the corresponding viewing directions. Gaussian Splatting optimizes the parameters involved in the scene rendering function described in the previous section. This includes the means and covariances of the Gaussian densities, their opacities, and the color function parametrized by spherical harmonics. Denoting by $\theta$ these parameters, the following reconstruction loss is used

$$\min_\theta \frac{1}{m} \sum_{k=1}^m \mathcal{L}(I_k, \hat{I}_{d_k,\theta}), \tag{2}$$

where $\hat{I}_{d_k,\theta}$ is the rendered frame of the scene in the direction $d_k$, as in Eq. (1), by using the parameters $\theta$, and $\mathcal{L}$ is a combination of $\ell_1$ and SSIM loss functions (Kerbl et al., 2023).

### 3.2 UPLIFTING OF 2D FEATURE MAPS INTO 3D

Given a set of $m$ 2D training frames and the corresponding 3D scene obtained by Gaussian Splatting, our goal is to compute generic features $f_i$ in $\mathbb{R}^c$ for each Gaussian $i$, which would be effective for solving future downstream tasks, *e.g.*, high-resolution semantic segmentation for new frames of the scene, or robot navigation. In other words, $f_i$ can be seen as an extension of the color function $c_i$, even though, for simplicity, we do not consider view-dependent features in this work.

A natural approach is to consider a pre-trained vision model that provides 2D feature maps for each of the $m$ frames used in Gaussian splatting, and then devise a technique to *uplift* these 2D feature maps into 3D. This uplifting principle can also be directly applied to semantic masks instead of generic features, as demonstrated in Section 5. Interestingly, once the features $f_i$ are computed for each Gaussian $i$, it is possible to *render* two-dimensional feature maps for any new view, at a resolution that can be much higher than the feature maps computed for the $m$ training frames.

**Uplifting with simple aggregation.** We construct uplifted features for each 3D Gaussian of the 3D Gaussian Splatting scene as a weighted average of 2D features from all frames. Each 2D feature $F_{d,p}$ from a frame at a given viewing direction $d$ and pixel $p$ contributes to the feature $f_i$ by a factor proportional to the rendering weight $w_i(d, p)$, if the Gaussian $i$ belongs to the ordered set $\mathcal{S}_{d,p}$ associated to the view/pixel pair $(d, p)$. The resulting features are then normalized to maintain the same order of magnitude as the original 2D features, thus resulting in the following simple equation:

$$f_i = \sum_{d=1}^{m} \sum_p \bar{w}_i(d, p) F_{d,p} \quad \text{with} \quad \bar{w}_i(d, p) = \frac{\mathbb{1}_{i \in \mathcal{S}_{d,p}} w_i(d, p)}{\sum_{d=1}^{m} \sum_p \mathbb{1}_{i \in \mathcal{S}_{d,p}} w_i(d, p)}, \tag{3}$$

where $\mathbb{1}_{i \in \mathcal{S}_{d,p}}$ is equal to 1 if the Gaussian $i$ belongs to $\mathcal{S}_{d,p}$ and 0 otherwise. We can interpret this equation as a normalized version of the transposed rendering operation over the $m$ viewing directions. More precisely, the rendering of any view-independent collection of features $\mathbf{f} = (f_i)$ attached to the $n$ Gaussians into the $m$ training frames can be represented as a linear operator $W$ acting on the collection $\mathbf{f}$ and returning a collection of 2D feature maps $\hat{\mathbf{F}} = (\hat{F}_{d,p})$, see (4) below. Here, the matrix $W$ consists of all rendering weights $\mathbb{1}_{i \in \mathcal{S}_{d,p}} w_i(d, p)$ at row $(d, p)$ and column $i$, and $\hat{\mathbf{F}}$ is a 2D matrix containing all (flattened) 2D feature maps generated for all cameras poses, with $\hat{\mathbf{F}}_{d,p}$ pointing to the feature of pixel $p$ viewed from camera pose $d$. Similarly, the uplifting expression introduced in Eq. (3) can be expressed in terms of the transpose of $W$ and a diagonal matrix $D$ of size $m$ representing the normalization factor and whose diagonal elements are obtained by summing over the rows of $W$ as in Eq. (5) below:

|  |  |  |  |
|---|---|---|---|
| Rendering to $m$ frames | | Uplifting from $m$ frames | |
| $\hat{\mathbf{F}} = W\mathbf{f},$ | (4) | $\mathbf{f} = D^{-1} W^\top \mathbf{F}.$ | (5) |

It is important to note that $W$ and $D$ are not explicitly constructed. Instead, they are computed by calling the forward rendering function for Gaussian Splatting and replacing the color vectors by the feature vectors. All these operations are performed within the CUDA rendering process. The procedure in (5), illustrated in Figure 1, bears similarity with the one from Chen et al. (2024) for uplifting 2D binary masks to a 3D Gaussian splatting scene. In their method, uplifted masks are thresholded to create 3D binary masks that can be rendered into different 2D frames. Such a thresholding operation would not be appropriate for uplifting generic features such as those from DINOv2. Moreover, unlike in Eq. (3) and (5), Chen et al. (2024) propose to normalize their uplifted masks based on the total count of view/pixel pairs $(d, p)$ contributing to the mask of a Gaussian $i$, i.e. $\sum_{d=1}^{m} \sum_p \mathbb{1}_{i \in \mathcal{S}_{d,p}}$, without taking the rendering weight $w_i(d, p)$ into account. Consequently, the uplifted features tend to have larger values for large, opaque Gaussians, making the rendering of these features more likely to ignore details provided by smaller and more transparent Gaussians.

**Connection with optimization-based inverse rendering.** An alternative approach to uplifting 2D features $\mathbf{F}$ is to minimize a reconstruction objective $\mathcal{L}(\mathbf{f})$, where the goal is to find uplifted features $\mathbf{f}$ whose rendering closely matches the original 2D features $\mathbf{F}$ (Tschernezki et al., 2022; Kerr et al., 2023; Zuo et al., 2024). A natural choice is to minimize the mean squared error between the 2D features $\mathbf{F}$ and the rendered ones $\hat{\mathbf{F}}$ as defined by Eq. (4):

$$\min_{\mathbf{f}} \mathcal{L}(\mathbf{f}) := \frac{1}{2} \| \mathbf{F} - W\mathbf{f} \|^2. \tag{6}$$

Such an approach requires using an optimization procedure which would be costly compared to the proposed uplifting method. Nevertheless, it is possible to interpret the proposed uplifting scheme in Eq. (5) as a single pre-conditioned gradient descent step on the reconstruction objective, starting from a $\mathbf{0}$ feature, i.e., $\mathbf{f} = -D^{-1} \nabla \mathcal{L}(\mathbf{0})$. In practice, we found that performing more iterations on the objective $\mathcal{L}(\mathbf{f})$ did not result in particular improvement of the quality of the features, thus suggesting that the cheaper scheme in Eq. (5) is already an effective approach to uplifting.

**Gaussian filtering** The normalization $\beta_i = \sum_{d=1}^{m} \sum_p \mathbb{1}_{i \in \mathcal{S}_{d,p}} w_i(d, p)$ serves as an estimator of the relative importance of each Gaussian in the scene. Therefore, it can be used as a criterion to prune the set of Gaussians for memory efficiency. In our experiments, we filter out half of the Gaussians based on $\beta_i$ and observe no qualitative nor quantitative degradation of the results. This approach is inspired by prior work on efficient Gaussian Splatting representation such as proposed by Fan et al. (2023) that also prunes Gaussians based on their contribution to each pixel in the training frames.

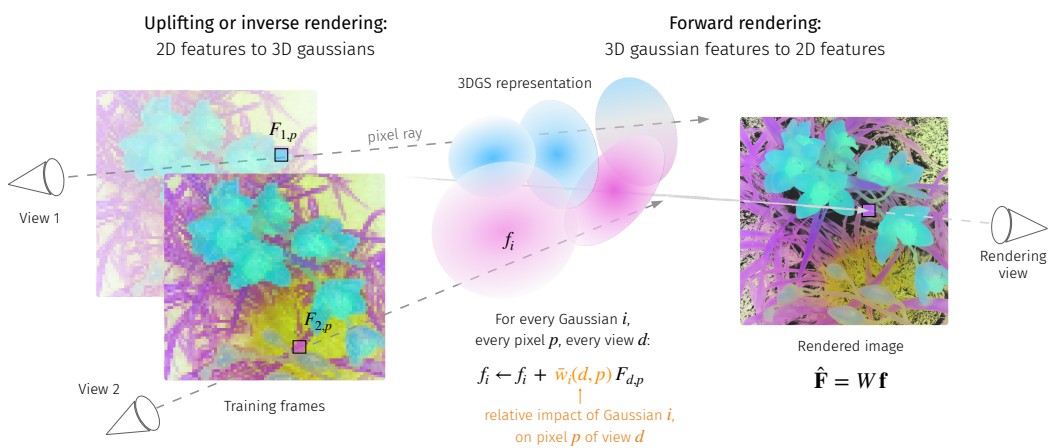

**Figure 1:** Illustration of the inverse and forward rendering. In the inverse rendering (or uplifting) phase, features $\mathbf{f}$ are created for each 3D Gaussian by aggregating coarse 2D features $\mathbf{F}$ over all viewing directions. For forward rendering, the 3D features $\mathbf{f}$ are projected on any given viewing direction as in regular Gaussian splatting. The rendering weight $\bar{w}_i(d, p)$ represents the relative influences of the Gaussian $i$ on pixel $p$, defined in Eq. (3).

### 3.3 ENRICHING FEATURES BY DIFFUSION ON GRAPHS

DINOv2 features have shown remarkable performance on semantic segmentation tasks with simple linear probing (Oquab et al., 2024), making them a good candidate to enrich features that lack such a property like CLIP (Wysoczańska et al., 2024; Zuo et al., 2024; Liu et al., 2023). Inspired by spectral clustering techniques (Shi & Malik, 2000; Kondor & Lafferty, 2002; Belkin & Niyogi, 2001), and aligning with the goals of recent work on 2D segmentation that improve CLIP features with DINO (Wysoczańska et al., 2024), we then propose to *diffuse* features that have been uplifted to 3D by leveraging pairwise similarities induced by DINOv2 while taking into account 3D geometry.

**Graph construction** From uplifted DINOv2 features $f$ in $\mathbb{R}^n$ we construct a graph whose nodes are given by the 3D Gaussians and whose edges, represented by a matrix $A$ of size $n \times n$, encode both the 3D Euclidean geometry between the nodes and the similarity between their DINOv2 features.

More precisely, each node $i$ is connected to its $k$ nearest neighbors $\mathcal{N}(i)$ as measured by the Euclidean distance between the centers of the 3D Gaussians. The edge weight between neighboring nodes $i$ and $j$ is given by a local feature similarity $S_f(f_i, f_j)$ between their DINOv2 features, typically a cosine similarity or an RBF kernel. For segmentation tasks, we prevent diffusion into the background by adding a node-wise unary regularization term $P(f_i)$, a similarity between node feature $f_i$ and some reference features $\bar{f}$. For details on $S_f$ and $P$ please refer to Appendix A.3.

$$A_{ij} = \mathbb{1}_{j \in \mathcal{N}(i)} \, S_f(f_i, f_j) \, P(f_i). \tag{7}$$

**Diffusion on the graph.** Given uplifted features $g_0$ in $\mathbb{R}^n$, which we would like to improve by using information encoded in $A$ (3D geometry or DINOv2 similarities, or both), we perform $T$ diffusion steps to construct a sequence of diffused features $(g_t)_{1 \le t \le T}$ defined as follows:

$$g_{t+1} = A\tilde{g}_t, \quad \tilde{g}_t = g_t / \|g_t\|_2, \tag{8}$$

which can be seen as performing a few steps of the power method, making $g_0$ closer to the dominant eigenspace of $A$. Note that depending on the downstream task, $g_0$ may be CLIP features, but it may also represent uplifted 2D segmentation masks provided by SAM.

## 4 FROM 3D UPLIFTING TO DOWNSTREAM TASKS

In this section, we describe our approach for uplifting DINOv2, SAM and CLIP models and evaluating the 3D features on two downstream tasks: segmentation and open-vocabulary object detection.

As in Sec. 3, we are given a set of 2D frames $I_1, \ldots, I_m$, with camera poses $d_1, \ldots, d_m$ and corresponding 3D scene obtained by the Gaussian Splatting method.

## 4.1 MULTIPLE-VIEW SEGMENTATION

We assume that a *foreground mask* of the object to be segmented is provided on the *reference frame* $I_1$. The foreground masks are either *scribbles* or a whole *reference mask* of the object, both of which define a set of *foreground pixels* $\mathcal{P}$. In the following, we present the proposed approaches for segmentation using SAM and DINOv2 features, based on both types of foreground masks.

**Multiple-view Segmentation with SAM.** SAM (Kirillov et al., 2023; Ravi et al., 2024) is a powerful model that can generate object segmentation masks from point prompts, on a single 2D image. Aggregating SAM 2D segmentation masks in 3D allows for cross-view consistency and improves single-view segmentation results. We proceed by generating 2D feature maps based on SAM segmentation masks of each training frame while only relying on the *foreground mask* for the reference frame $I_1$. The 2D feature maps are generated by constructing several sets of point prompts on each training frame which are then provided to SAM to obtain several segmentation masks. The point prompts are obtained using the *foreground mask* provided on the *reference frame* as described in Appendix A.1. Averaging the resulting segementation masks for each frame results in the final 2D SAM feature maps. These are then uplifted using the aggregation scheme in Sec. 3.2. Our final prediction is obtained by rendering the uplifted feature maps into the target frame.

**Multiple-view segmentation with DINOv2.** We construct 2D feature maps at the patch level using DINOv2 with registers (Darcet et al., 2024) and uplift them into a high resolution and fine-grained 3D semantic representation which is then used for segmentation. The 2D feature maps are constructed using a combination of a sliding windows mechanism and dimensionality reduction of the original DINOv2 features as described in Appendix A.2 and illustrated in Fig. 4 therein. This approach enhances the granularity of spatial representations by aggregating patch-level representations to form pixel-level features. The 2D feature maps from the $m$ training views are uplifted using Eq. (3) and the resulting 3D features are then re-projected into any viewing direction $d$ using Eq. (4) to compute rendered 2D features $(\hat{F}_{d,p})$. To obtain segmentation masks, we construct a predictor score $P(\hat{F}_{d,p})$ for a 2D pixel $p$ to belong to the foreground, based on its corresponding rendered feature. The score $P$ is obtained by comparing the rendered features $(\hat{F}_{d,p})$ with *foreground features* $\mathcal{F}_{ref} := (\hat{F}_{d_1,p})_{p \in \mathcal{P}}$ corresponding to the *foreground mask* computed on the *reference frame* $I_1$, see Appendix A.2. The final segmentation mask is then obtained by thresholding.

**Enhancing segmentation with DINOv2 using 3D graph diffusion.** DINOv2 provides generic visual features that do not explicitly include segmentation information, unlike models such as SAM that were specifically trained for such a task. Consequently, 2D projections of uplifted DINOv2 features might fail to separate distinct objects that have similar features. This challenge can be mitigated by incorporating 3D spatial information, which may help separate them.

To this end, we propose to leverage the graph diffusion process introduced in Section 3.3. We set the initial vector of weights $g_0$ in $\mathbb{R}^n$ of the graph diffusion algorithm to be a coarse estimation of the contribution of each Gaussian to the final segmentation mask. This initial weight vector is computed by uplifting the 2D *foreground mask* (either scribbles or a reference mask) from the *reference frame* into 3D using Eq. (3), normalizing and thresholding them (see appendix Sec. A.3). The nodes for which $g_0$ has a positive value define a set of anchor nodes $\mathcal{M}$ that are more likely to contribute to the foreground. We retain the last weight vector $g_T$ and render it into 2D for segmentation (Eq. (4)). The regularization term $P$ appearing in Eq. (7) is obtained by comparing the uplifted features with anchor features obtained using the *foreground mask* as described in the appendix.

## 4.2 OPEN-VOCABULARY OBJECT DETECTION

As in (Kerr et al., 2023; Qin et al., 2024; Zuo et al., 2024) we propose to uplift CLIP features (Radford et al., 2021) which are excellent for aligning images and text, and evaluate the uplifted features task of open-vocabulary detection (Kerr et al., 2023). As CLIP outputs only one feature vector per input image, a couple of extra steps are needed to distill CLIP into the 3D Gaussians.

**Construction of CLIP feature maps** We follow the common practice (Kerr et al., 2023; Zuo et al., 2024) of constructing multi-resolution CLIP 2D feature maps by querying CLIP on a grid of overlapping patches at different scales and aggregating the resulting representations. As in Zuo et al. (2024), rather than keeping the different representations separate, we choose to aggregate them with a simple average pooling. These multi-resolution CLIP features are uplifted into 3D using Eq. (3).

**Refinement with DINOv2 graph diffusion** We further refine those features with the diffusion procedure described in Sec. 3.3. To this end, DINOv2 features are also uplifted, and the similarity matrix is built as in Eq. (7), with no unary term $P$. The diffusion process locally aggregates CLIP features from Gaussians whose DINOv2 features are similar. This enhances the granularity of CLIP visual representations while remaining in the CLIP feature space, allowing for downstream applications with text queries. Wysoczańska et al. (2024) perform a similar procedure for 2D image segmentation, with only one step in the diffusion. This process can also be related to the pixel-alignment loss in (Zuo et al., 2024), as a diffusion step corresponds to a gradient step for that loss.

## 5 EXPERIMENTS

### 5.1 EXPERIMENT DETAILS

**3D scene training and pruning.** All scenes are trained using the original Gaussian Spatting implementation (Kerbl et al., 2023) with default hyperparameters. For memory efficiency, half of the Gaussians are filtered out based on their importance, as described in Sec. 3.2.

**2D vision models.** Our experiments are conducted using DINOv2's ViT-g with registers (Darcet et al., 2024), SAM (Kirillov et al., 2023), SAM 2 (Ravi et al., 2024) and the OpenCLIP ViT-B/16 model (Ilharco et al., 2021).

**Segmentation tasks.** We consider two segmentation tasks: i) Neural Volumetric Object Selection (NVOS, Ren et al. 2022), which is derived from the LLFF dataset (Mildenhall et al., 2019), and ii) SPIn-NeRF, which contains a subsets of scenes from NeRF-related datasets (Knapitsch et al., 2017; Mildenhall et al., 2019; 2021; Yen-Chen et al., 2022; Fridovich-Keil et al., 2022). The NVOS dataset consists of forward-facing sequences in which one frame is labeled with a segmentation mask and another one is labeled with scribbles to be used as reference. SPIn-NeRF contains both forward-facing and 360-degree scenes, in which all frames are labeled with segmentation masks, and the standard evaluation protocol uses the segmentation mask from the first frame as reference to label the subsequent frames.

**Open-vocabulary object detection** We evaluate on the LERF Localization dataset (Kerr et al., 2023) consisting of complex in-the-wild scenes. We report our results on the extended evaluated task introduced by LangSplat (Qin et al., 2024) containing additional challenging localization samples.

**Evaluation and hyperparameter tuning.** Our segmentation results are averaged over 3 independent runs. Segmentation with 3D SAM masks requires setting a threshold for foreground/background pixel assignment, and optionally choosing one of the three masks proposed by SAM (representing different possible segmentations of the object of interest). Segmentation with DINOv2 uses graph diffusion with RBF kernels as the similarities $P$ and $S_f$ and therefore needs three hyperparameters: the 2 bandwidths of the RBF kernels, and the threshold for foreground/background pixel assignment.

For SPIn-NeRF, all hyperparameters are chosen based on the IoU for the available reference mask. For NVOS, only reference scribbles are provided, hence i) for SAM/SAM2, only one mask is generated, and the threshold for segmentation is fixed for all scenes for SAM and automatically chosen using Li iterative Minimum Cross Entropy method (Li & Lee, 1993) for SAM 2, ii) for DINOv2 we predict a SAM mask based on the scribbles of the reference frame, and choose the hyperparameters maximizing the IoU with this SAM mask. This is consistent with a scenario where the user, here SAM, would choose hyperparameters based on visual inspection on one of the frames.

For the LERF Localization task, graph diffusion is run with $P = 1$ and an RBF kernel for $S_f$, with a set of different bandwidths. The resulting feature maps are automatically selected based on the relevancy score with the text prompt. This aligns with the semantic level selection process of LERF (Kerr et al., 2023) and LangSplat (Kerbl et al., 2023).

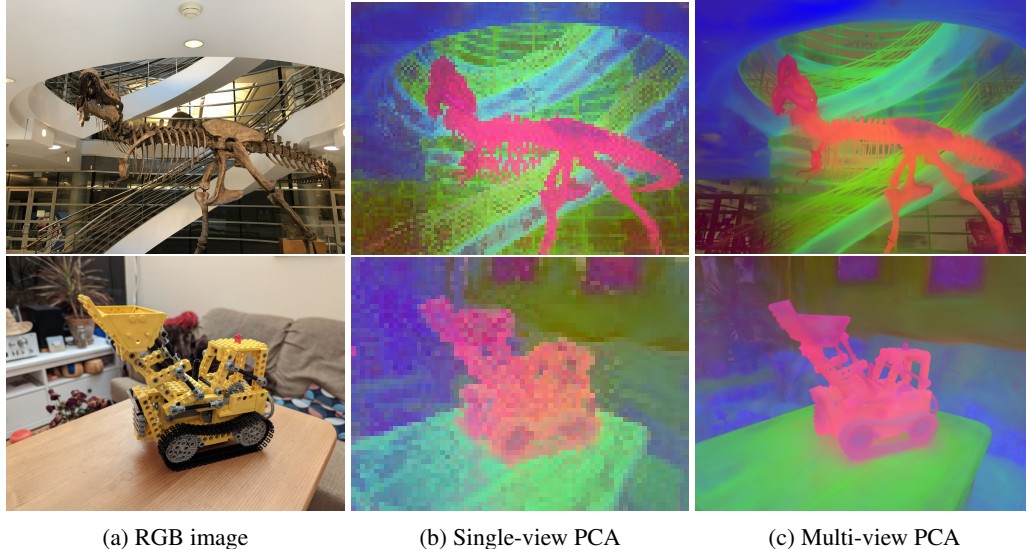

(a) RGB image        (b) Single-view PCA        (c) Multi-view PCA

Figure 2: **PCA visualizations.** The DINOv2 patch-level representations (middle) predicted from the RGB images (left) are aggregated into highly detailed 3D representations (right) using Eq. 3.

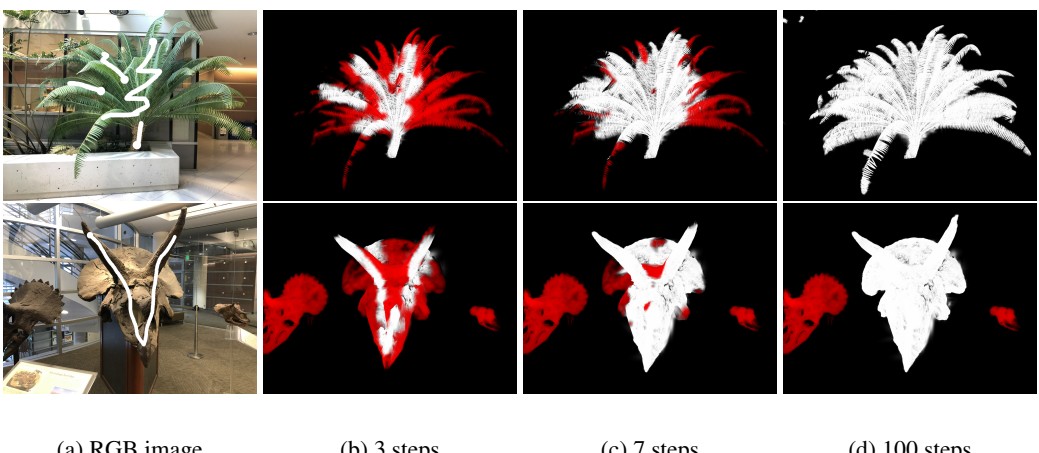

(a) RGB image      (b) 3 steps      (c) 7 steps      (d) 100 steps

Figure 3: **Illustration of the diffusion process.** 2D projection of the weight vector $g_t$ (white) and unary regularization term (red) at different diffusion steps $t$. The diffusion process allows filtering out unwanted objects that have similar features to the object of interest (such as the two smaller skulls on *horns*, bottom-row), but are disconnected in space. The regularization term (red background) prevents leakage from the object to the rest of the scene (such as through the *fern*'s trunk, top-row).

## 5.2 QUALITATIVE RESULTS

**DINOv2 feature uplifting.** First, we illustrate the effectiveness of our simple uplifting approach. Figure 2 shows the first three PCA components (one channel per component) over DINOv2's patch embeddings. The coarse patch-level representations from every view (middle) are aggregated using Eq. 5 to form a highly detailed 3D semantic representation, and reprojected into 2D (right) using Eq. 4. The aggregation is very fast, as it is directly implemented in the Gaussian Splatting CUDA-based rendering process. The procedure takes about 1.5ms per view and can be parallelized across the feature dimension. The first principal component (encoded in the red channel) mostly captures the foreground object, and the subsequent ones allow refining the foreground representations and delivering a detailed background. In the appendix, we provide additional comparative visualizations of our learned 3D features (Fig. 8) and of 3D segmentation for scene editing (Fig. 7).

| Geometry only | Single view | | Uplifting | | Uplifting + Graph diffusion |
|---|---|---|---|---|---|
| Reference mask | DINOv2 | SAM2 | DINOv2 | SAM2 | DINOv2 |
| 80.4 | 88.3 | 90.5 | 90.8 | 93.7 | 92.8 |

Table 1: Segmentation (IoU) on SPIn-NeRF (Mirzaei et al., 2023). We compare purely geometrical reference mask uplifting and reprojection and single-view prediction, feature/mask uplifting or graph diffusion leveraging DINOv2 or SAM2.

| | MVSeg | SA3D-TRF | SA3D-GS | SAGA | OmniSeg3D | LUDVIG (Ours) | | |
|---|---|---|---|---|---|---|---|---|
| 3D representation: | | TensoRF | GS | GA | NeRF | | GS | |
| Uplifting: | | SAM | SAM | SAM | SAM | DINOv2 | SAM | SAM2 |
| NVOS | - | 90.3 | 92.2 | 92.6 | 91.7 | 92.4 | 91.3 | 91.2 |
| SPIn-NeRF | 90.9 | 93.7 | 93.2 | 93.4 | 94.3 | 92.8 | 93.7 | 93.7 |

Table 2: Segmentation (IoU) on NVOS (Ren et al., 2022) and SPIn-NeRF (Mirzaei et al., 2023).

**Graph diffusion.** Figure 3 illustrates the effectiveness of the diffusion process. In the Fern scene, diffusion progressively spreads through the branches to their extremities and the regularization (red background) prevents it from leaking beyond the trunk. As illustrated with the case of Horns, diffusion filters out unwanted objects that are similar to the object of interest (here the two skulls on the side). The graph nodes are initialized with the reference scribbles and the diffusion spreads through the object of interest and stop at its borders. The regularization sets a constraint that prevents leakage, even after a large number of iterations. This is also illustrated in Appendix Figure 6 for the Flower and Trex scenes: diffusion rapidly spreads, with near-full coverage after only 5 steps, before reaching all the much smaller Gaussians on the border, allowing for a refined segmentation.

## 5.3 SEGMENTATION RESULTS

In this section, we quantitatively evaluate the segmentation task on NVOS (Ren et al., 2022) and SPin-NeRF (Mirzaei et al., 2023). We evaluate segmentation based on SAM and SAM2 mask uplifting, and on DINOv2 feature uplifting combined with graph diffusion. We compare our segmentation results to the current state of the art: SA3D (Cen et al., 2023c), SA3D-GS (Cen et al., 2023b), SAGA (Cen et al., 2023a), OmniSeg3D (Ying et al., 2024). All these methods are specifically designed for uplifting the 2D segmentation masks produced by SAM into 3D using gradient-based optimization of a projection loss. We also report results from NVOS (Ren et al., 2022) and MVSeg (Yen-Chen et al., 2022), who respectively introduced the NVOS and SPIn-NeRF datasets.

**Results.** Table 2 reports the average IoU across all scenes for the two datasets. Per-scene results can be found in Appendix Tables 4 and 5. Our results comparable to the state of the art while not relying on gradient-based optimization. Surprisingly, our segmentation with DINOv2 using graph diffusion also gives results on par with models leveraging SAM masks. Compared to SAM, DINOv2 better captures complex objects, but sometimes also capture some background noise. This can be seen in Appendix Figure 5 with the example of Trex: while SAM misses out the end of the tail as well as the end of the ribs, DINOv2 captures the whole Trex, but also captures part of the stairs behind. Our lower segmentation results compared to OmniSeg's can be partly attributed to poor Gaussian Splatting reconstruction of highly specular scenes such as the Fork, in which semi-transparent Gaussians floating over the object try to represent reflections or surface effects that are difficult to capture with standard rasterization techniques (Jiang et al., 2024).

**Ablation study.** We compare our segmentation protocol using DINOv2 and SAM2 to multiple simpler variants. More precisely, we evaluate i) a purely geometrical variant that reprojects the reference mask on the other views, without using SAM2 or DINOv2, ii) single-view segmentation in 2D based on SAM2 or DINOv2 2D predictions, iii) uplifting DINOv2 features or SAM2 masks into 3D then rendering them for segmentation, and iv) segmenting using graph diffusion over DINOv2 3D feature similarities. Results are reported in Table 1, and per-scene IoU as well as a detailed analysis can be found in Appendix Table 6 and Sec. B.1. We observe that the purely geometrical approach works well on the forward-facing scenes and fails on 360-degree scenes. The single-view

|  | LERF Loc. dataset | | Extended LERF Loc. dataset | | |
|---|---|---|---|---|---|
|  | LERF | FMGS | LERF | LangSplat | LUDVIG |
| ramen | 62.5 | 90.0 | 62.0 | 73.2 | 77.5 |
| figurines | 87.2 | 89.7 | 75.0 | 80.4 | 78.6 |
| teatime | 96.9 | 93.8 | 84.8 | 88.1 | 94.9 |
| waldo_kitchen | 85.2 | 92.6 | 72.7 | 95.5 | 86.4 |
| **overall** | 83.0 | 91.5 | 73.6 | 84.3 | 84.4 |
| **average time (mins)** | 45 | 100 | 45 | 105 | 9 |

Table 3: **LERF Localization** We evaluate on the more challenging dataset introduced by LangSplat (Qin et al., 2024) and report results from LERF (Kerr et al., 2023) and FMGS (Zuo et al., 2024) on the original dataset.

variant performs reasonably well on average but, the low resolution of patch-level representations (illustrated in Fig. 2) lead to a coarser segmentation. 3D uplifting considerably boosts results compared to single-view approaches, and introducing 3D spatial information through 3D graph diffusion further enhances results on the more challenging 360-degree scenes.

## 5.4 OPEN-VOCABULARY OBJECT DETECTION

Table 3 presents results on the open-vocabulary localization task. The reported average running times, include feature map generation and 3D feature training whenever relevant. For reference, we report the results of LERF and FMGS (Zuo et al., 2024) on the original version of the LERF localisation dataset introduced in Kerr et al. (2023). We also report LERF Kerr et al. (2023) and LangSplat (Qin et al., 2024) on the extended and more challenging version of the LERF localisation dataset introduced by LangSplat (Qin et al., 2024), on which LERF incurs a significant drop in performance. LUDVIG performs on par with LangSplat and outperforms LERF on the extended LERF localisation dataset while being significantly faster than all methods (around 10 times faster).

A more thorough analysis on running times can be found in appendix Sec. B.2. Additionally, Appendix Sec. C.3 provides illustrations of the impact of the diffusion process (Fig. 10), and comparative visualizations of localization heatmaps with LangSplat and LERF (Fig. 11).

## 6 CONCLUDING REMARKS AND LIMITATIONS

**Learning-free uplifting.** In this work, we introduce a simple yet effective aggregation mechanism for transferring 2D visual representations into 3D, bypassing the traditional optimization-based approach. The aggregation builds upon already trained Gaussian Splatting representations and is implemented within the CUDA rendering process, making 2D-to-3D uplifting as fast as 3D-to-2D rendering. Note however that the quality of learned 3D features is bound by that of the 3D scene reconstruction. Reconstruction by Gaussian Splatting is notoriously challenging when dealing with, *e.g.*, highly specular scenes (Jiang et al., 2024; Yang et al., 2024), blurred images Zhao et al. (2024); Lee et al. (2024) or high-frequency regions (Ye et al., 2024b; Zhang et al., 2024). In such scenarios, learning 3D features *along with* 3D Gaussian Splatting reconstruction may lead to improved scene geometry, opening promising perspectives for future work.

**Graph diffusion.** Our proposed graph diffusion process allows injecting the rich DINOv2 representations to refine arbitrary features such as segmentation masks or CLIP embeddings. Our CLIP feature refinement builds upon prior works using DINO features as a regularization (Kerr et al., 2023; Zuo et al., 2024), while alleviating the computational overhead associated with joint gradient-based optimization of CLIP and DINO features. However, it does rely on the adequate choice of bandwidth hyperparameter(s) when defining the similarity graph. In this work, these hyperparameters are chosen based on IoU with a SAM-predicted mask for segmentation, and based on the relevancy score with the text prompt for object localization. While automatic, this decision process requires multiple evaluations of a success criterion with different candidate bandwidth values.

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

# Appendix

## A    USING LUDVIG FOR DOWNSTREAM TASKS

In this section, we describe our approach for uplifting DINOv2, SAM and CLIP models and evaluating the 3D features on two downstream tasks: segmentation and open-vocabulary object detection.

As in Sec. 3, we are given a set of 2D frames $I_1, \ldots, I_m$, with camera poses $d_1, \ldots, d_m$ and corresponding 3D scene obtained by the Gaussian Splatting method, which can be used to uplift 2D features from the $m$ frames to 3D.

**Multiple-view segmentation.**    For this task, we assume that a *foreground mask* of the object to be segmented is provided on a *reference frame* taken to be the first frame $I_1$. We consider two types of foreground masks: either *scribbles* or a whole *reference mask* of the object, both of which define a set of *foreground pixels* $\mathcal{P}$. In the following subsections, we present the proposed approaches for segmentation using SAM and DINOv2 features, based on both types of foreground masks.

### A.1    MULTIPLE-VIEW SEGMENTATION WITH SAM

SAM (Kirillov et al., 2023; Ravi et al., 2024) is a powerful image segmentation model, that can generate object segmentation masks from point prompts on a single 2D image. Aggregating SAM 2D segmentation masks in 3D allows for cross-view consistency and improves single-view segmentation results. In order to leverage SAM, we propose a simple mechanism for generating SAM 2D features for each frame from a *foreground mask* in the *reference frame*.

**Construction of 2D feature maps.**    The key idea is to generate point prompts on each training frame from the *foreground mask* provided on the *reference frame*. To this end, we perform an uplifting of the *foreground mask* (Eq. (3)) and re-project it on all frames (Eq. (4)). This results in 2D scalar maps that we further normalize by their average value. A higher values indicates the presence of the target object. For each frame with camera pose $d$, we retain a subset of pixels $\mathcal{P}_d$ with values higher than a threshold fixed for all scenes and select point prompts for SAM from this subset. Finally, we compute 2D segmentation masks for each frame using SAM by randomly selecting 3 points prompts from $\mathcal{P}_d$, repeating the operation 10 times and averaging the resulting masks for each view to obtain the final 2D SAM feature maps.

**Segmentation with uplifted SAM masks.**    The 2D segmentation masks generated by SAM are uplifted using the aggregation scheme described in Sec. 3.2. Our final prediction is obtained by rendering the uplifted feature maps into the target frame.

### A.2    MULTIPLE-VIEW SEGMENTATION WITH DINOV2

DINOv2 (Oquab et al., 2024) is a self-supervised vision model recognized for its generalization capabilities. In this work, we aggregate the patch-level representations produced by DINOv2 with registers (Darcet et al., 2024) into a high resolution and fine-grained 3D semantic representation.

**Construction of 2D feature maps.**    We construct the 2D feature maps using a combination of a sliding windows mechanism and dimensionality reduction of the original DINOv2 features. Specifically, we i) extract DINOv2 patch-level representations across multiple overlapping crops of the training images, ii) apply dimensionality reduction over the set of all patch embeddings, ii) upsample and aggregate the dimensionality-reduced patch embeddings to obtain pixel-level features for each image. The process is illustrated in Figure 4. This approach enhances the granularity of spatial representations by aggregating patch-level representations to form pixel-level features.

**Segmentation with uplifted DINOv2 features.**    The 2D feature maps from the $m$ training views are uplifted using Eq. (3) and the resulting 3D features are then re-projected into any viewing direction $d$ using Eq. (4) to compute rendered 2D features $(\hat{F}_{d,p})$. To obtain segmentation masks, we construct a score $P(\hat{F}_{d,p})$ for a 2D pixel $p$ to belong to the foreground, based on its corresponding rendered feature. More precisely, $P$ relies on the rendered *foreground features* $\mathcal{F}_{ref} := (\hat{F}_{d_1,p})_{p \in \mathcal{P}}$

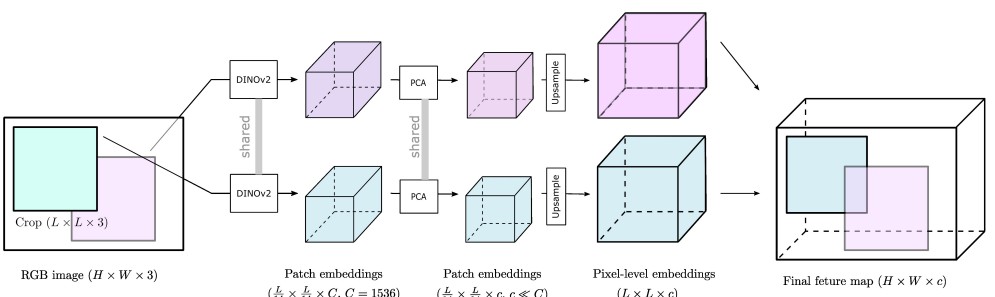

Figure 4: Sliding windows for construction of DINOv2 feature maps.

corresponding to the *foreground mask* computed on the *reference frame* $I_1$. We propose two approaches for constructing $P$. The first one is a simple approach that sets $P(\hat{F}_{d,p}) = \mathcal{S}_F(\hat{F}_{d,p}, \bar{F})$ where $\bar{F}$ is the average over foreground features $\mathcal{F}_{ref}$, and $\mathcal{S}_F$ is defined based on the cosine similarity. The second approach is more discriminative and first trains a logistic regression model $P$ on all rendered 2D features of the reference frame, so that the foreground features $\mathcal{F}_{ref}$ are assigned a positive label. Then $P(\hat{F}_{d,p})$ gives the probability that a pixel $p$ belongs to the foreground. The final mask is then obtained by thresholding.

Experimentally, the second approach is extremely efficient when the set of *foreground pixels* $\mathcal{P}$ covers the whole object to segment, so that $P$ captures all relevant features. This is the case when a whole *reference mask* of the object is provided. When the *foreground pixels* $\mathcal{P}$ does not cover the whole object, as with scribbles, $P$ can be discriminative to parts of the object that are not covered by $\mathcal{P}$. Therefore, we rely on the second approach for tasks where a reference mask is provided, and use the simpler first approach when only scribbles serve as reference.

### A.3 ENHANCING SEGMENTATION WITH DINOV2 USING 3D GRAPH DIFFUSION

DINOv2 provides generic visual features that do not explicitly include information for segmentation, unlike models such as SAM that were specifically trained for such a task. Consequently, using the 2D projections of uplifted DINOv2 features, as proposed in Sec. A.2, might fail to separate different objects that happen to have similar features while still being distinct entities. This challenge can be mitigated by incorporating 3D spatial information in which the objects are more likely to be well-separated. To this end, we propose to leverage the graph diffusion process introduced in Section 3.3.

For this task, the initial vector of weights $g_0 \in \mathbb{R}^n$ representing a coarse estimation of the contribution of each Gaussian to the segmentation mask. We retain the last weight vector $g_T$ and render it into 2D for segmentation (Eq. (4)). Below, we describe the initialization of the weight vector $g_0$ and the construction of the adjacency matrix $A$.

**Initialization of the weight vector.** The initial weight vector $g_0$ is computed by uplifting the 2D *foreground mask* (either scribbles or a reference mask) from the *reference frame* into 3D using Eq. (3), normalizing the 3D mask by its mean value over all nodes and setting to zero all values below a fixed threshold. The nodes for which $g_0$ has a positive value define a set of anchor nodes $\mathcal{M}$ that are more likely to contribute to the foreground. The resulting weight vector is a coarse estimation of how much each Gaussian contributes to a rendered 2D segmentation mask.

**Construction of the graph edges.** We define $S_f$ based on the cosine similarity between features and choose a global unary regularization term $P(f_i)$ on each node $i$ to encourages similarity between the uplifted node feature $f_i$ and those belonging to the foreground. More precisely, $P$ is defined using a similar approach as in Sec. A.2: either as a similarity $P(f_i) = \mathcal{S}_f(f_i, \bar{f})$ with the averaged feature $\bar{f}$ over the anchor nodes $\mathcal{M}$ (in the case when scribbles are provided), or as a logistic regression model trained on the uplifted features, so that anchor nodes' features are assigned a positive label (in the case when a full foreground mask is available). The local term $S_f$, typically a cosine similarity, allows diffusion to neighbors that have similar features while the unary term prevents leakage to background nodes during diffusion by encouraging closeness to the foreground features and allows using an arbitrary number of diffusion steps.

| | MVSeg | SA3D-GS | SAGA | OmniSeg3D | LUDVIG (Ours) | | |
|---|---|---|---|---|---|---|---|
| 3D representation: | NeRF | GS | GS | NeRF | | GS | |
| Uplifting: | | SAM | SAM | SAM | DINOv2 | SAM | SAM2 |
| Orchids | 92.7 | 84.7 | - | 92.3 | 92.6 | 91.9 | 90.7 |
| Leaves | 94.9 | 97.2 | - | 96.0 | 93.9 | 96.4 | 96.4 |
| Fern | 94.3 | 96.7 | - | 97.5 | 95.6 | 96.8 | 96.7 |
| Room | 95.6 | 93.7 | - | 97.9 | 94.7 | 96.5 | 96.6 |
| Horns | 92.8 | 95.3 | - | 91.5 | 94.4 | 92.3 | 94.9 |
| Fortress | 97.7 | 98.1 | - | 97.9 | 97.6 | 98.3 | 98.3 |
| Fork | 87.9 | 87.9 | - | 90.4 | 81.6 | 87.1 | 86.8 |
| Pinecone | 93.4 | 91.6 | - | 92.1 | 90.1 | 90.8 | 90.8 |
| Truck | 85.2 | 94.8 | - | 96.1 | 94.8 | 94.3 | 92.6 |
| Lego | 74.9 | 92.0 | - | 90.8 | 93.2 | 92.8 | 92.9 |
| Average | 90.9 | 93.2 | 93.4 | 94.3 | 92.8 | 93.7 | 93.7 |

Table 4: Segmentation (IoU) on SPIn-NeRF (Mirzaei et al., 2023) with DINOv2, SAM and SAM2.

| | Fern | Flower | Fortress | HornsC | HornsL | Leaves | Orchids | Trex | Average |
|---|---|---|---|---|---|---|---|---|---|
| NVOS | - | - | - | - | - | - | - | - | 70.1 |
| SA3D | 82.9 | 94.6 | 98.3 | 96.2 | 90.2 | 93.2 | 85.5 | 82.0 | 90.3 |
| OmniSeg3D | 82.7 | 95.3 | 98.5 | 97.7 | 95.6 | 92.7 | 84.0 | 87.4 | 91.7 |
| SA3D-GS | - | - | - | - | - | - | - | - | 92.2 |
| SAGA | - | - | - | - | - | - | - | - | 92.6 |
| Ours-DINOv2 | 84.4 | 96.3 | 95.3 | 95.4 | 93.4 | 95.9 | 92.1 | 86.4 | 92.4 |
| Ours-SAM | 85.5 | 97.6 | 98.1 | 97.9 | 94.1 | 96.4 | 73.1 | 88.0 | 91.3 |
| Ours-SAM2 | 84.8 | 97.2 | 98.3 | 97.7 | 92.4 | 96.9 | 73.0 | 89.0 | 91.2 |

Table 5: Segmentation (IoU) on NVOS (Ren et al., 2022) with DINOv2, SAM and SAM2.

# B   ADDITIONAL RESULTS

## B.1   PER-SCENE SEGMENTATION RESULTS

In this section, we present per-scene segmentation results on NVOS and SPIn-NeRF in Tables 4, 5 and 6, along with an extended analysis of these results..

**Segmentation on SPIn-NeRF.** We report our segmentation results for the SPin-NeRF dataset (Mirzaei et al., 2023) in Table 4. Our results are comparable to the state of the art while not relying on optimization-based approaches. Surprisingly, our segmentation with DINOv2 using graph diffusion also gives results on par with models leveraging SAM masks. Our lower segmentation results compared to OmniSeg's can be partly attributed to poor Gaussian Splatting reconstruction of highly specular scenes such as the Fork, in which semi-transparent Gaussians floating over the object try to represent reflections or surface effects that are difficult to capture with standard rasterization techniques (Jiang et al., 2024).

**Segmentation on NVOS.** We report our segmentation results for the NVOS dataset (Ren et al., 2022) in Table 5. Our results are comparable to those obtained by prior work. Again, DINOv2 performs surprisingly well while not having been trained on billions of labeled images like SAM. Compared to SAM, DINOv2 better captures complex objects, but sometimes also capture some background noise. This can be seen in Appendix Figure 5 with the example of Trex: while SAM misses out the end of the tail as well as the end of the ribs, DINOv2 captures the whole Trex, but also captures part of the stairs behind. Visualisations of Orchids in Appendix Figure 5 also explain the lower performance of SAM on this scene: the two orchids SAM is missing are not covered by the positive scribbles, which makes the task ambiguous.

**Ablation study** In Table 6, we compare our segmentation protocol using DINOv2 and SAM2 to multiple simpler variants. More precisely, we evaluate i) a purely geometrical variant that does not use SAM2 or DINOv2, ii) single-view segmentation in 2D based on SAM2 or DINOv2 2D predic-

| Model: | Geometry only | Single view | | Uplifting | | Graph diffusion |
|---|---|---|---|---|---|---|
| | Reference mask | DINOv2 | SAM2 | DINOv2 | SAM2 | DINOv2 |
| Orchids | 80.9 | 91.4 | 79.2 | 91.7 | 90.7 | 92.6 |
| Leaves | 94.8 | 89.3 | 96.6 | 94.3 | 96.4 | 93.9 |
| Fern | 95.5 | 94.4 | 96.7 | 96.7 | 96.7 | 95.6 |
| Room | 85.7 | 94.5 | 96.3 | 97.1 | 96.6 | 94.7 |
| Horns | 90.4 | 90.7 | 92.7 | 93.1 | 94.9 | 94.4 |
| Fortress | 95.4 | 96.8 | 97.8 | 98.7 | 98.3 | 97.6 |
| Fork | 66.3 | 85.6 | 77.2 | 88.4 | 86.8 | 81.6 |
| Pinecone | 58.8 | 92.9 | 90.3 | 86.7 | 90.8 | 90.1 |
| Truck | 60.0 | 86.2 | 89.3 | 88.8 | 92.6 | 94.8 |
| Lego | 77.2 | 63.5 | 89.1 | 72.4 | 92.9 | 93.2 |
| Average | 80.4 | 88.3 | 90.5 | 90.8 | 93.7 | 92.8 |

Table 6: **Segmentation (IoU) on SPIn-NeRF (Mirzaei et al., 2023)**. We compare purely geometrical reference mask uplifting and reprojection and single-view prediction, feature/mask uplifting or graph diffusion leveraging DINOv2 or SAM2.

tions, iii) uplifting DINOv2 features or SAM2 masks into 3D then rendering them for segmentation, as described in Sec. A.1 and A.2, and iv) segmenting using graph diffusion over DINOv2 3D feature similarities.

The purely geometrical approach works well on the forward-facing LLFF scenes (Orchids to Fortress). In these scenes, the reference mask is accurately uplifted and reprojected as the viewing direction changes only a little between each frame. However, it fails on the 360-degree scenes (Fork, Pinecone, Truck, Lego). This points to a suboptimal 3D reconstruction of the scene, likely due to overfitting on the limited numbers of available training views (Chung et al., 2024).

The single-view variants use a similar process for constructing the features and using them for segmentation as in Sec. A.1 and A.2 but without uplifting and rendering. It improves from a purely geometrical approach and performs reasonably well on average, the foreground being well isolated from the rest of the scene. However, as illustrated in Figure 2, the semantic features are at a much lower resolution than those resulting from 3D uplifting, leading to a coarser segmentation.

3D uplifting considerably boosts results compared to single-view approaches. However, performing segmentation in 2D based on the uplifted DINOv2 features does not benefit from the 3D spatial information and typically fails on the 360-degree scenes (Pinecone, Truck and Lego) which have higher variability between frames from different views. Introducing 3D spatial information through 3D graph diffusion results in a boosted performance on these scenes.

### B.2 RUNTIME ANALYSES

The total reported times can be divided between pre-uplifting, uplifting and post-uplifting. These steps are detailed below. Experiments for LUDVIG are run on a GPU A6000 ADA.

**Pre-uplifting.** This step includes 2D feature map generations. The time this step takes depends on the backbone model, on the number of training images and on the number of calls to the model per image. The total time ranges from a few seconds up to an hour for models such as LangSplat (Qin et al., 2024), that queries SAM over a grid of points on the image at various resolutions to generate full image segmentation masks. This process takes 24s/image on a GPU 6000 ADA and amounts to an average of 80 minutes for the evaluated scenes In our experiments, the feature generation step takes from 1 to 5 minutes.

**Uplifting.** For LUDVIG, uplifting takes 1.3ms per feature dimension for an image of size $480 \times 640$. For example, uplifting 100 training images with a feature dimension of 40 takes 5 seconds. Gradient-based optimization requires approximately additional time, where represents the number of gradient steps, typically ranging from 3,000 to 30,000 for 3D feature distillation (Kerr et al., 2023; Qin et al., 2024; Zuo et al., 2024). Gradient-based optimization can still be very fast for low-dimensional features such as SAM masks (can take as little as a few seconds, as reported

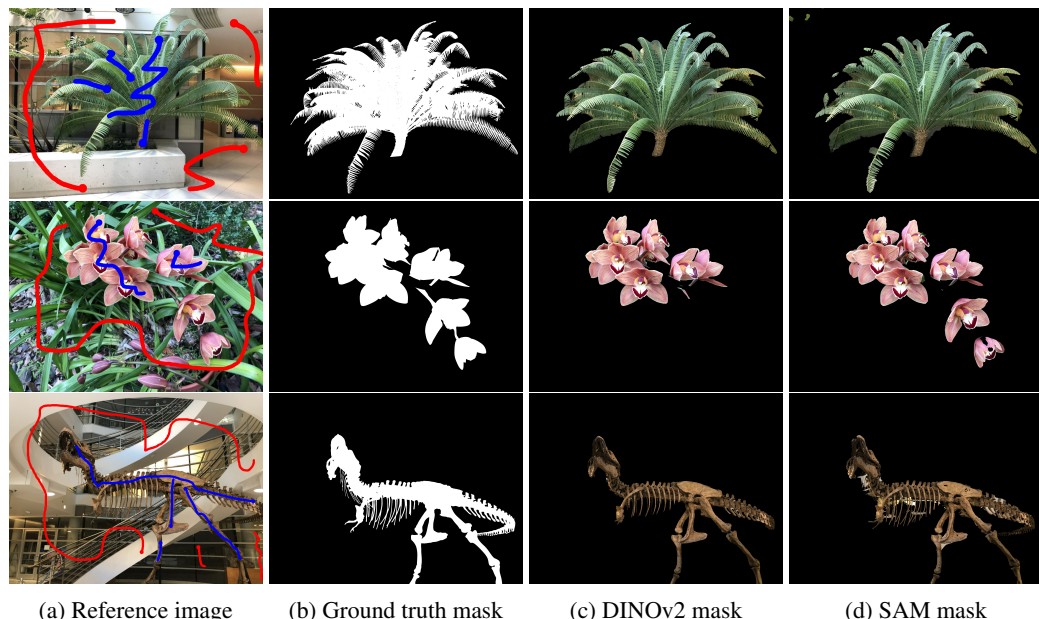

| (a) Reference image | (b) Ground truth mask | (c) DINOv2 mask | (d) SAM mask |

Figure 5: Segmentation results on NVOS (Ren et al., 2022) with DINOv2 and SAM.

by SA3D-GS (Cen et al., 2023b)) or dimensionality-reduced features (LangSplat (Qin et al., 2024) trains an autoencoder to reduce the CLIP feature dimension from 512 to 3 and runs for 25 minutes). However, optimization becomes intractable for high-dimensional features such as CLIP and DINO; FMGS (Zuo et al., 2024) relies on an efficient multi-resolution hash embedding of the scene; however, their total training time still amounts to 1.4 hours.

**Post-uplifting.** After uplifting, LUDVIG leverages graph diffusion using pairwise DINOv2 feature similarities for segmentation tasks as well as for CLIP feature refinement. This refinement can be seen as a proxy for regularization losses used in prior works when jointly training CLIP and DINOv2 features. Graph diffusion first requires querying the nearest neighbors for each node, which is linear in the number of Gaussians and takes about 1 minute with 1,000,000 Gaussians. This can be further optimized by using approximate nearest neighbor search algorithms (Wang et al., 2021). The diffusion then takes less than 1 second for 1D features such as segmentation masks, and up to 30 seconds for higher-dimensional features such as CLIP. Therefore, graph diffusion comes as a small overhead to the total running time.

## C  ADDITIONAL VISUALISATIONS

### C.1  SEGMENTATION TASKS

**Segmentation on NVOS.** Figure 5 shows our segmentation masks from SAM and DINOv2 for the three most challenging scenes of the NVOS dataset: Fern, Orchids and Trex.

**Diffusion process.** Figure 6 illustrates different steps of the diffusion process for Fern, Leaves, Flower and Trex from the NVOS (Ren et al., 2022) dataset. Starting from the reference scribbles, the diffusion rapidly spreads through the large neighboring Gaussians. Covering the entire object takes more time for complex structures such as Fern, or for masks with disconnected components such as Orchids. As illustrated in the case of Flower, the last diffusion steps allow spreading to the smaller Gaussians on the flowers' edges, yielding a refined segmentation mask. For Trex, the parts being reached the latest are the head and tail. Their features are further away from the reference features (defined as the average feature over 3D reference scribbles), and therefore the regularization for diffusion is stronger in these regions. Overall once the object has been fully covered, the regularization is very effective at preventing leakage, which allows diffusion to run for an arbitrary number of steps.

**Scene editing.**     Figure 7 shows comparative visualizations of scene editing with N3F (Tschernezki et al., 2022) and LUDVIG. For rendering the edited RGB image, N3F sets to zero the occupancy for all 3D points belonging to the object. For LUDVIG, we remove all Gaussians pertaining to the 3D semantic mask resulting from graph diffusion. We observe that the regions behind to segmented object are much smoother for LUDVIG than for N3F. Regions unseen from any viewpoint are black for LUDVIG (no gaussians) and result in a background partially hallucinated by NeRF for N3F.

## C.2    Visualizations of uplifted DINOv2 features

**Visual comparisons with N3F.**     Figure 8 show a comparison of LUDVIG's 3D DINOv2 features with learned 3D DINO features of N3D (Tschernezki et al., 2022). Their figures are taken from their work. The notable differences are a more fine-grained reconstruction of the background for the trex and horns, and overall smoother features across all scenes.

**Comparison to GaussianEditor's uplifting.**     Figure 9 shows a qualitative comparison of our proposed aggregation with the one introduced by GaussianEditor Fan et al. (2023) (see Sec. 3. The visualizations illustrate that GaussianEditor's aggregation fails to assign the right semantics to large gaussians, which is particularly visible in scenes with high specularity such as Room. This showcases the importance of defining 3D features as *convex combinations* of 2D pixel features.

## C.3    Visualization of CLIP features and localization task

In this section, we present illustrations of the impact of the diffusion process (Figure 10), and comparative visualizations of localization heatmaps with LangSplat and LERF (Figure 11).

**Impact of DINOv2-guided graph diffusion for CLIP feature refinement.**     Figure 10 shows PCA visualizations of uplifted CLIP features before and after refinement with graph diffusion as well as DINOv2 features used to define edge weights. Graph diffusion allows transferring DINOv2 visual representations into the CLIP feature space, which is well illustrated with the top example: after diffusion, the two figurines on the foreground acquire different semantics. The diffusion process also yields more homogeneous features for a given object, as illustrated with the ramen bowl in the middle, or the table at the bottom. Globally, graph diffusion greatly enhances the semantic coherence and granularity within the scene.

**Qualitative comparison of open-vocabulary objet localization.**     Figure 11 illustrates open-vocabulary object localization with LERF (Kerr et al., 2023), LangSplat (Qin et al., 2024) and LUDVIG. Both LangSplat and LUDVIG correctly localize all four example objects. For queries such as the chopsticks, LangSplat's localization is more precise, as the CLIP features are constructed by generating full image segmentation masks with SAM. This process is computationally expensive, as constructing a full segmentation mask requires querying SAM over a grid of points on the image and takes about 23s for a single image (on a GPU A6000 ADA), which amounts to an average of 80 minutes for a scene from the LERF dataset. However, it yields coherent instance-level CLIP representations, which is desirable for downstream segmentation tasks.

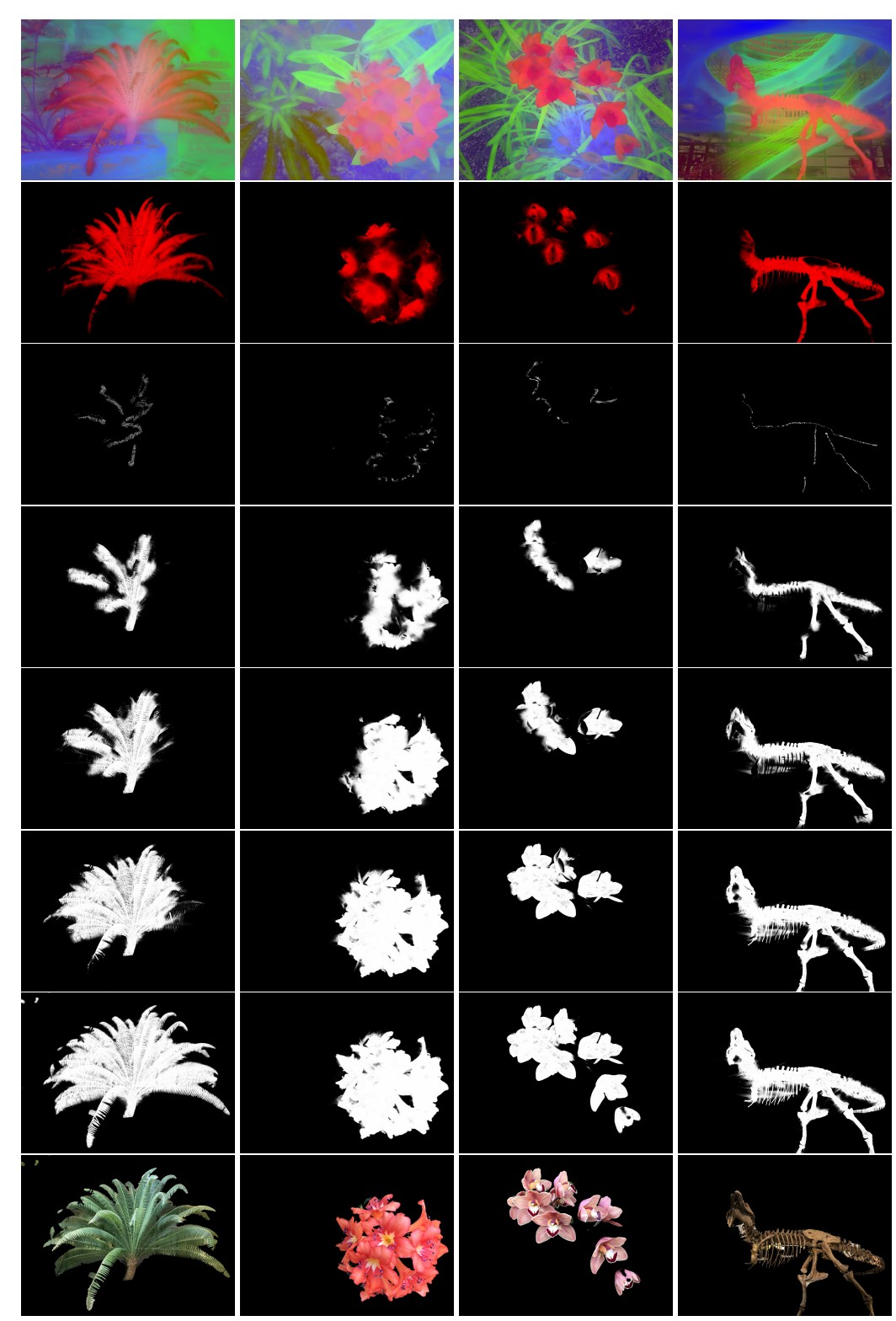

Figure 6: **Illustration of the graph diffusion process.** 2D projections of i) first three PCA components of DINOv2 3D features, ii) unary regularization term (red), iii) weight vector $g_t$ at timesteps $t \in \{0, 3, 5, 10, 100\}$, iv) RGB segmentation obtained using a mask based on the 2D projection of $g_{100}$.

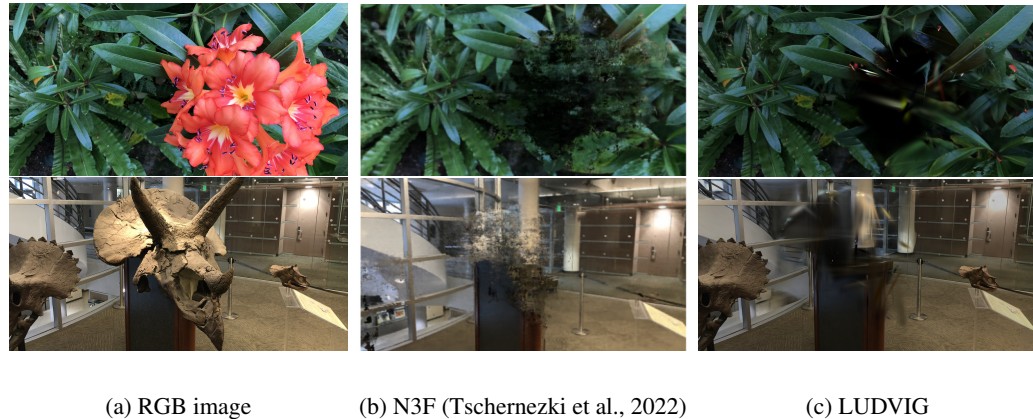

(a) RGB image      (b) N3F (Tschernezki et al., 2022)      (c) LUDVIG

Figure 7: **Scene editing.** 3D segmentation, removal and rendering for LUDVIF and N3F (Tschernezki et al., 2022). For N3F, figures are sourced from (Tschernezki et al., 2022).

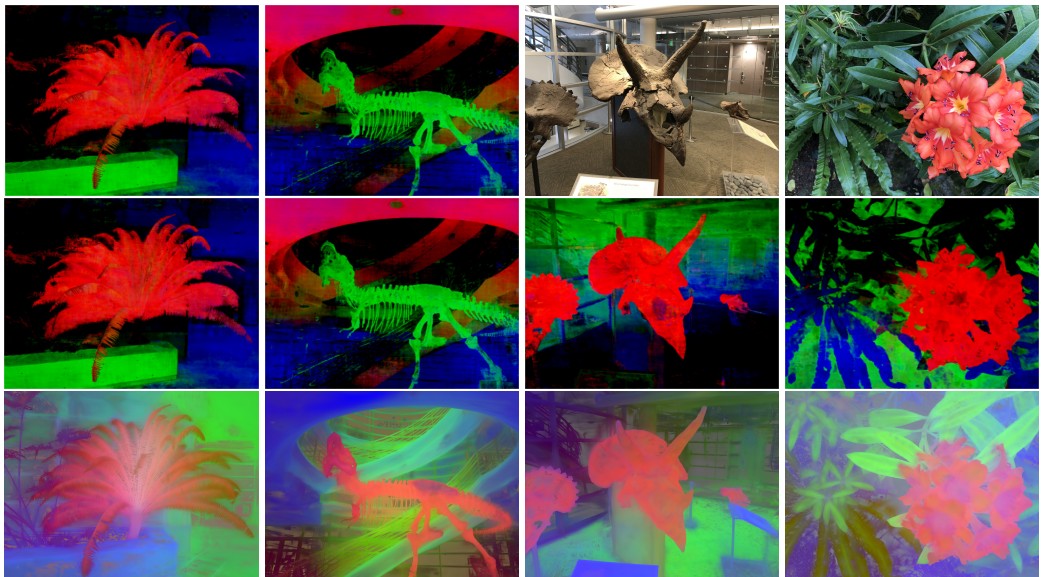

Figure 8: Comparison between LUDVIG's uplifted DINOv2 features (bottom) and N3F's (Tschernezki et al., 2022) learned DINO features (top). For N3F, figures are sourced from (Tschernezki et al., 2022).

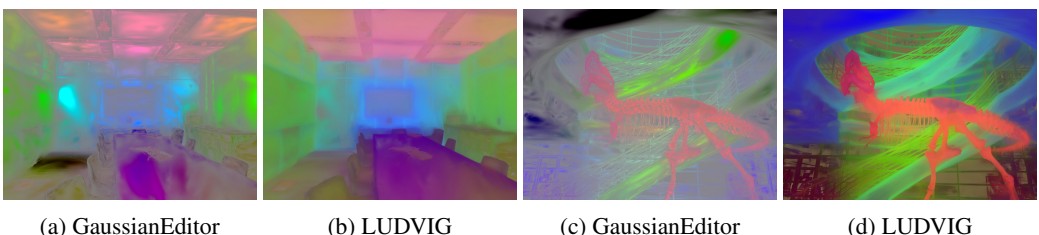

(a) GaussianEditor      (b) LUDVIG      (c) GaussianEditor      (d) LUDVIG

Figure 9: **Comparison to GaussianEditors's uplifting.** Comparison of PCA visualization of uplifted features between LUDVIG's and GaussianEditor's aggregation (Chen et al., 2024).

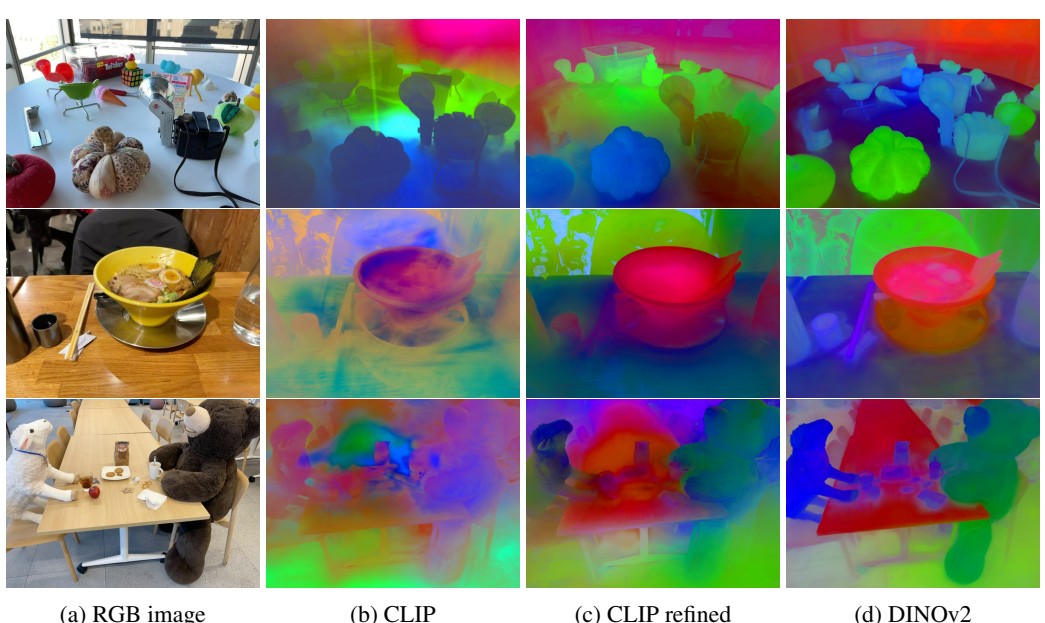

     (a) RGB image        (b) CLIP        (c) CLIP refined      (d) DINOv2

Figure 10: **CLIP, refined CLIP and DINOv2 features.** PCA visualizations of 3D CLIP features, 3D CLIP features refined using graph diffusion with DINOv2 similarities, and DINOv2 features.

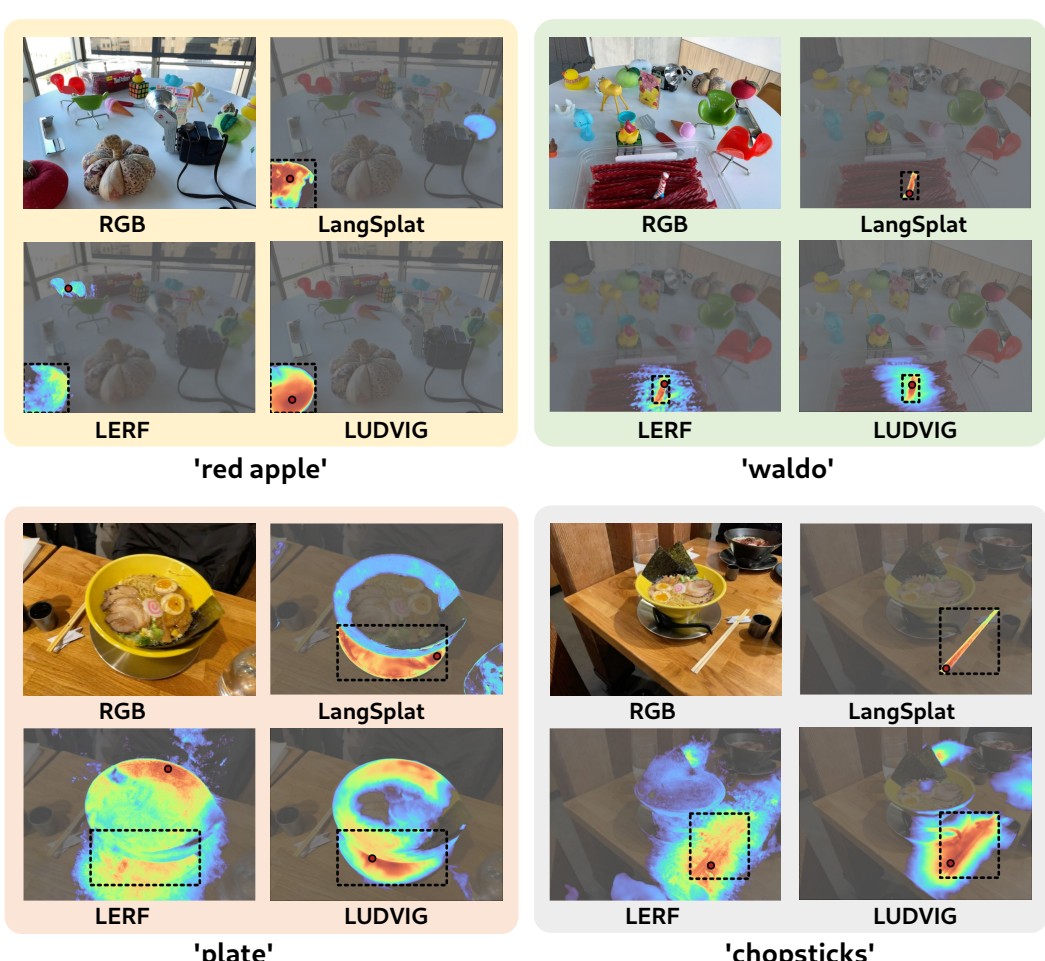

Figure 11: **Qualitative comparisons of open-vocabulary 3D object localization on the LERF dataset.** The red points are the model predictions and the black dashed bounding boxes denote the annotations. This figure is sourced and adapted from LangSplat's website.

