# OpenReview forum: "LUDVIG: Learning-free Uplifting of 2D Visual Features to Gaussian Splatting Scenes"
_ICLR.cc/2025/Conference — Submitted to ICLR 2025_

### Official Review · Reviewer_QRFz · 2024-11-03

**Soundness:** 2
**Presentation:** 3
**Contribution:** 1
**Rating:** 3
**Confidence:** 2

**Summary:**

This paper presents a method that lifts 2D features into the 3D representation space of 3DGS. With the lifted features, the method can perform segmentation of objects, and other applications. The paper tested features of SAM and DINO. For DINO, a graph diffusion step is performed to enhance the results.

**Strengths:**

1. Previous approaches adopt optimizations, while this method is training-free and maybe fast.
2. The presentation is clear.

**Weaknesses:**

1. The method needs additional post-processing steps such as graph diffusion.
2. The paper is not fully finished. There is no conclusion section.

**Questions:**

1. The lifting algorithm is very simple, which is a weighted average of features collected from multiple views. This reviewer had an experience on object segmentation based on NeRF. I feel that the aggregated features may suffer from noises. For example, there may be errors in camera parameters, which can reduce the effectiveness of the aggregation. I suggest the authors to add more analysis on this problem (In fact, I do not know how to evaluate this, and expect the authors to provide more insights).

2. The method is probably highly affected by the effectiveness of the feature extraction backbone. I am wondering whether the 2D segmentation masks obtained by the backbones applied to the images are similar to those of the lifted 3DGS projected to the 2D. Please provide this additional comparison.

3. I do not see connection between the proposed method and optimization-based approach in lines beginning from 205. Maybe this paragraph discusses the difference between the two kinds of approaches. I think a title of “Difference with optimization-based inverse rendering” may be more appropriate.

4. The method needs a graph diffusion to further enhance its results. Why do you need this step? Please provide the results w/ and w/o this step.

5. Please check the paper and make sure all sections are completed.

Overall, I think the accuracy of this method may be limited due to the noises in the features themselves caused by, e.g., misalignment between features of different views. The method is fast, but there is no argument supporting the necessity of this method at the cost of accuracy.

---

> ### Author Response · Authors · 2024-11-22
> **Answer to Reviewer QRFz**
>
> We thank the reviewer for their valuable insights on evaluating the quality of our 3D representations, including their robustness to noise and the comparison with more direct approaches, as well as their observation regarding the lack of concluding remarks. We hope our responses adequately address their concerns.
>
> > Robustness of aggregated features to noise
>
> We thank the reviewer for their insightful comment. Gaussian Splatting indeed requires accurate camera pose estimation, which we agree can be challenging in settings with, e.g., noisy data, limited view overlap or ambiguous feature matches due to a lack of texture, occlusions, or motion blur. However, we would like to politely point to the fact that this is beyond the scope of our paper and of every work uplifting 2D features in 3D. A large body of work focuses on that exact problem such as [1,2,3]. We are happy to add a discussion about that aspect in the supplementary if the reviewer finds it useful.
>
> *References*
>
> [1] Darmon et al. (2024). “Robust Gaussian Splatting”
>
> [2] Matsuki et al. (2024). “Gaussian Splatting SLAM”
>
> [3] Lee et al. (2024). “Deblurring 3D Gaussian Splatting”
>
> > *How does the uplifting compare with directly predicting segmentation masks in 2D with the backbone models?*
>
> Please refer to Table 3, ‘Single view’ column, that exactly answers this question. It corresponds to the segmentation results obtained by predicting on a single image. Please note that we did our best to report the strongest possible results for those baselines to ensure the comparison is as fair as possible. In particular, those reported for SAM are much higher than what reported in prior works such as 3DGS.
>
> > Impact of graph diffusion
>
> The graph diffusion allows isolating a specific object instance in the scene using 3D information and similarity with the query. This is important for DINOv2, whose features are similar for objects of the same semantic category and cannot be directly used for instance segmentation, unlike  SAM which was trained for segmentation. In our latest experiments on open-vocabulary object localization, graph diffusion using DINOv2 feature similarities also proved a powerful approach for refining 3D CLIP features.
>
> > Results with and without graph diffusion.
>
> The comparison with and without graph diffusion is provided in Table 3 under the name “Graph diffusion” and “Uplifting” respectively. We will clarify this in the revised version by changing the name “Graph diffusion” to “Uplifting + Graph diffusion” and we apologize if this has caused confusion.
>
> > “Connection with optimization-based inverse rendering”
>
> For optimization-based inverse rendering, a natural objective to minimize is the one introduced in eq 6 *w.r.t.* the 3D features $f$. A standard algorithm for optimizing such objective is to update the 3D features $f$ using several steps of gradient descent or pre-conditioned gradient descent. Pre-conditioning here means multiplying the gradient by a symmetric positive matrix (for instance the diagonal matrix $D^{-1}$ that we consider). When initializing the features to $0$ and updating them using  a single preconditioned gradient step one recovers exactly our proposed uplifting rule in eq 5.
>
> > Concluding remarks
>
> We will add concluding remarks that emphasize the impact of our contributions and address the limitations of our work, including its dependence on some hyperparameters such as the choice of bandwidth for diffusion, and its reliance on accurately reconstructed 3D scenes.

---

> ### Comment · Reviewer_QRFz · 2024-11-27
>
> Thanks for the feedback. The basic concern is that I cannot get the point why this method is effective, especially the uplifting scheme itself. If the uplifting algorithm is not your main contribution, I think you can tune its importance down in the paper.

---

> > ### Author Response · Authors · 2024-11-28
> >
> > > Thanks for the feedback. The basic concern is that I cannot get the point why this method is effective, especially the uplifting scheme itself. If the uplifting algorithm is not your main contribution, I think you can tune its importance down in the paper.
> >
> > There is an ongoing discussion with Rev. *sBLh* and *jxL3* where we further demonstrate the effectiveness of our method: an order of magnitude faster on open-vocabulary localization, with accuracy on par with SOTA methods.
> >
> > We would greatly appreciate knowing if you see any areas of disagreement with the arguments we have presented.

---

### Official Review · Reviewer_jxL3 · 2024-11-03

**Soundness:** 3
**Presentation:** 3
**Contribution:** 3
**Rating:** 6
**Confidence:** 3

**Summary:**

The paper introduces LUDVIG, a learning-free method designed to uplift 2D visual features or semantic masks to 3D Gaussian Splatting scenes. LUDVIG avoids traditional, iterative optimization processes by using a simple aggregation technique, achieving comparable segmentation quality to state-of-the-art methods. The approach is applied effectively to features from both SAM and DINOv2, demonstrating high segmentation performance and computational efficiency across complex 3D scenarios.

**Strengths:**

1. The learning-free feature uplifting method is both simple and effective, achieving strong results without training.
2. Experiments with SAM and DINOv2 demonstrate the method’s efficiency, yielding performance comparable to training-based approaches.
3. High Computational Efficiency: LUDVIG bypasses the costly and time-consuming optimization steps typical in 3D reconstruction methods, making it highly efficient.
4. Versatile Input Adaptability: The proposed method adapts seamlessly to various input types, such as features from SAM and DINOv2, showcasing robust performance across diverse feature sources.

**Weaknesses:**

1. While the method is straightforward, it relies on hand-crafted processes, such as the segmentation score calculation and the graph diffusion process. These manual strategies may raise concerns about robustness, particularly in complex, real-world scenarios.
2. Certain sections, like Sec. 4.2, are challenging to follow. For example, the construction of 2D feature maps from DINOv2 is not clearly outlined. Including diagrams or visual aids could greatly enhance understanding and clarify complex steps.
3. Lack of qualitative comparison results with other baselines.
4. The proposed method is primarily limited to foreground segmentation, whereas other feature lifting approaches support additional tasks, such as instance segmentation, 3D segmentation, and downstream applications like scene manipulation.

**Questions:**

1. As in weakness 2, can you explain the process of feature map construction?
2. How is the time efficiency of the proposed method compared to other training-based methods?
3. As in weakness 4, Can the proposed method performed on other tasks and how well is it?

---

> ### Author Response · Authors · 2024-11-22
> **Answer to Reviewer jxL3 (1/2)**
>
> We thank the reviewer for their thorough analysis of our work and results, as well as their suggestions for improving the clarity of our method section, extending our experimental evaluation to other tasks, and providing both qualitative comparisons and efficiency analyses. We hope the responses below address their questions and suggestions.
>
> > Performance on other tasks
>
> We qualitatively evaluated our approach on 3D segmentation and applications in scene editing. We will include these visualizations in the revised version. Additionally, we conducted an evaluation on object detection using CLIP. Inspired by prior works leveraging DINOv2 as regularization (LERF [1], FMGS [2]), we jointly uplifted CLIP and DINOv2 features and refined the 3D CLIP features using graph diffusion, aggregating them based on the cosine similarity between DINOv2 representations. The results, presented in the table below, show that our method is competitive with prior works while being significantly faster
>
> | Test Scene | LERF [1]	 | LangSplat [3]	| FMGS [2]	 | Ours (no diffusion) | Ours (diffusion)
> |------------------|---------------------|------------------|-------------------|------|---------
> | ramen    	|     62.5  	|    73.2 	 |       90.0 	 |  	71.8	  |   	77.5	|
> | figurines 	|    79.5	 |   80.4  	|         89.7   	 |	42.9	 |       	78.6	|
> | teatime   	|    93.8	 |   88.1	|         93.8   	 |       	89.8	 |	94.9	|
>  | waldo_kitchen |    81.5 	|   95.5 	|      92.6          |    	81.8    	 |   	86.4	|
>  | overall	 |    80.3 	|   84.3	 	|       91.5     	| 	     71.6   	 |	84.4	|
>  | average time (minutes) |    	 45  |    105 	|      100        	|  	7  	| 	9    |
>
> > Efficiency compared to other training-based methods
>
>  We thank the reviewer for this remark, this is indeed lacking in our submission. For our latest experiments on object localization with CLIP, we included a comparison of training times with other methods. For this task, our feature generation, uplifting and graph diffusion take an average of 9 minutes and reach results on par with prior works while being almost 10 times faster. We will include this analysis in our revised version.
>
> The total reported times can be divided between pre-uplifting, uplifting and post-uplifting. More precisely,
> 1.  **Pre-uplifting** includes 2D feature map generations. The time this step takes depends on the backbone model, on the number of training images and on the number of calls to the model per image. The total time ranges from a few seconds up to an hour for models such as LangSplat [3], that queries SAM over a grid of points on the image at various resolutions to generate full image segmentation masks. In our experiments, this step takes from 1 to 5 minutes.
> 2. **Uplifting**: this takes 1.3ms per feature dimension for an image of size 480 $\times$ 640. For example, uplifting 100 training images with a feature dimension of 40 takes 5 seconds.  Gradient-based optimization requires approximately $n_{\text{steps}}$ additional time, where $n_{\text{steps}}$ represents the number of gradient steps, typically ranging from 3,000 to 30,000 for 3D feature distillation. Gradient-based optimization can still be very fast for low-dimensional features such as SAM masks (can take as little as a few seconds, as reported by SA3D-GS [4]) or dimensionality-reduced features (LangSplat trains an autoencoder to reduce the CLIP feature dimension from 512 to 3 and runs for 25 minutes). However, optimization becomes intractable for high-dimensional features such as CLIP and DINO; FMGS [2] relies on an efficient multi-resolution hash embedding of the scene; however, their total training time still amounts to 1.4 hours.
> 3.  **Post-uplifting**: after uplifting, LUDVIG leverages graph diffusion using pairwise DINOv2 feature similarities for segmentation tasks as well as for CLIP feature refinement. This refinement can be seen as a proxy for regularization losses used in prior works when jointly training CLIP and DINOv2 features. Graph diffusion first requires querying the $k$ nearest neighbors for each node, which is linear in the number of Gaussians and takes about 1 minute with 1,000,000 Gaussians and $k=200$. This can be further optimized by using approximate nearest neighbor search algorithms [5]. The diffusion then takes less than 1 second for 1D features such as segmentation masks, and up to 30 seconds for higher-dimensional features such as CLIP. Therefore, graph diffusion comes as a small overhead to the total running time.
>
> *References*
>
> [1] Kerr et al. (2023). “LERF: Language-Embedded Radiance Fields”
>
> [2] Zuo et al. (2024). “FMGS: Foundation Model Embedded 3D Gaussian Splatting for Holistic 3D Scene Understanding”
>
> [3] Qin et al. (2024). “LangSplat: 3D Language Gaussian Splatting“
>
> [4] Cen et al. (2024). "Segment Anything in 3D with Radiance Fields"
>
> [5] Wang et al. (2021). “A Comprehensive Survey and Experimental Comparison of Graph-Based Approximate Nearest Neighbor Search“

---

> > ### Author Response · Authors · 2024-11-22
> > **Answer to Reviewer jxL3 (2/2)**
> >
> > > Construction of 2D feature maps from DINOv2.
> >
> > We thank the reviewer for their feedback and will include an illustration of the 2D feature map construction in our revised version. The steps for constructing these feature maps are as follows:
> > 1. For each view, predict DINOv2 over a set of overlapping square crops ($L \times L$) from the RGB image, yielding patch-level representations of size $D \times \frac{L}{14}\times\frac{L}{14}$ with $D=1536$ for ViT-g.
> > 2. Apply PCA to the set of all patch embeddings from all crops, reducing $D$ to, e.g., 40 components, which results in patch-level representations of size $40 \times \frac{L}{14}\times \frac{L}{14}$.
> > 3. Bilinearly upsample patch-level representations to the original resolution of the crop and aggregate the resulting representations of size $40\times L \times L$. Since the crops are overlapping, the aggregated image-level representation of pixel $p$ is an average of features from the crops overlapping $p$.
> >
> > This approach is similar to the one used to generate multi-resolution CLIP image representations. The revised version will integrate a more general formulation embracing both use cases, and we will include this more detailed textual description in the supplementary.
> >
> > > Lack of qualitative comparison results with other baselines.
> >
> > We thank the reviewer for this remark, and will ensure that the revised version includes qualitative comparisons with prior works of our learned features, segmentation masks, and heatmaps of object localization.

---

> > > ### Comment · Reviewer_jxL3 · 2024-11-26
> > >
> > > Thank you for your efforts. All of my concerns and questions have been thoroughly addressed.

---

> > > > ### Author Response · Authors · 2024-11-28
> > > >
> > > > We sincerely appreciate your feedback. If you feel that our rebuttal or the revised version has adequately addressed your concerns and believe the paper is now suitable for acceptance, we would be truly grateful if you could consider adjusting your score accordingly.

---

### Official Review · Reviewer_N4ZX · 2024-11-04

**Soundness:** 3
**Presentation:** 3
**Contribution:** 2
**Rating:** 5
**Confidence:** 3

**Summary:**

This submission proposes a simple yet effective scheme of uplifting visual features from 3D scenes represented by Gaussian splitting. In contrast to the previous approaches, the proposed method enables direct 3D segmentation without iterative optimization, while achieves comparable to the SotAs.

**Strengths:**

+ The proposed scheme of connecting per-pixel 2D features and Gaussians are simple and intuitive.

+ The segmentation can be directly done without iterative optimization on a trained Gaussian.

+ The treatment on incorporating with DINOv2 feature into segmentation is nice, as it induces comparable results with the variant using the more tailored-for SAM.

**Weaknesses:**

- There is neither limitation/failure nor future work discussion in the submission, what is the boarder impact of the work for the community?

- The submission lacks report on running time.

**Questions:**

The graph diffusion (Eq.(7)) is similar to power method for computing the dominant eigenvector of a matrix. While it is not guaranteed to converge to the dominant eigenvector, I guess there is a high probability. Is it possible to draw some theoretical connection to, say, spectral clustering methods, and therefore give some insight of the converging point of g_t?

In figure 2, the rolling stair are separated from the ceiling in the multi-view case while merged to the latter in the single-view case, could the authors provide some explanation on such behavior?

In Sec. 4.2, the DINOv2 features seem to be of coarse scale (predicted in patches), would the recent works on feature super-resolution (e.g., Feature-up in ICLR 2024 by Fu et al.) help?

---

> ### Author Response · Authors · 2024-11-22
> **Answer to Reviewer N4ZX (1/2)**
>
> We thank the reviewer for their thorough analysis of our work and results, their suggestions for improvement, and their insightful questions regarding the inspiration behind the graph diffusion process. We hope the responses below address their remarks and questions.
>
> > Discussion of limitation, future work and broader impact
>
> Thank you for this relevant remark. We will include a section discussing the limitations and failure cases of our work in our revised version. One limitation of our work is indeed its reliance on some hyperparameters for the downstream tasks and for the diffusion process.  Another is our dependence on the quality of Gaussian Splatting reconstruction.
> The broader impact would be the ability to seamlessly transition from 2D to 3D for arbitrary representations, without having to define a specific loss and tune optimization hyperparameters. Our latest experiments also showed that graph diffusion can be used for feature refinement, serving as a proxy to the local regularization losses such as the pixel-alignment loss from FMGS [1].
>
> > *The submission lacks report on running time*
>
> We thank the reviewer for this remark, this is indeed lacking in our submission. For our latest experiments on object localization with CLIP, we included a comparison of training times with other methods. For this task, our feature generation, uplifting and graph diffusion take an average of 9 minutes and reach results on par with prior works while being almost 10 times faster. We will include this analysis in our revised version.
>
> The total reported times can be divided between pre-uplifting, uplifting and post-uplifting. More precisely,
> 1.  **Pre-uplifting** includes 2D feature map generations. The time this step takes depends on the backbone model, on the number of training images and on the number of calls to the model per image. The total time ranges from a few seconds up to an hour for models such as LangSplat [2], that queries SAM over a grid of points on the image at various resolutions to generate full image segmentation masks. In our experiments, this step takes from 1 to 5 minutes.
> 2. **Uplifting**: this takes 1.3ms per feature dimension for an image of size 480 $\times$ 640. For example, uplifting 100 training images with a feature dimension of 40 takes 5 seconds.  Gradient-based optimization requires approximately $n_{\text{steps}}$ additional time, where $n_{\text{steps}}$ represents the number of gradient steps, typically ranging from 3,000 to 30,000 for 3D feature distillation. Gradient-based optimization can still be very fast for low-dimensional features such as SAM masks (can take as little as a few seconds, as reported by SA3D-GS [3]) or dimensionality-reduced features (LangSplat trains an autoencoder to reduce the CLIP feature dimension from 512 to 3 and runs for 25 minutes). However, optimization becomes intractable for high-dimensional features such as CLIP and DINO; FMGS [1] relies on an efficient multi-resolution hash embedding of the scene; however, their total training time still amounts to 1.4 hours.
> 3.  **Post-uplifting**: after uplifting, LUDVIG leverages graph diffusion using pairwise DINOv2 feature similarities for segmentation tasks as well as for CLIP feature refinement. This refinement can be seen as a proxy for regularization losses used in prior works when jointly training CLIP and DINOv2 features. Graph diffusion first requires querying the $k$ nearest neighbors for each node, which is linear in the number of Gaussians and takes about 1 minute with 1,000,000 Gaussians and $k=200$. This can be further optimized by using approximate nearest neighbor search algorithms [4]. The diffusion then takes less than 1 second for 1D features such as segmentation masks, and up to 30 seconds for higher-dimensional features such as CLIP. Therefore, graph diffusion comes as a small overhead to the total running time.
>
> *References*
>
> [1] Zuo et al. (2024). “FMGS: Foundation Model Embedded 3D Gaussian Splatting for Holistic 3D Scene Understanding”
>
> [2] Qin et al. (2024). “LangSplat: 3D Language Gaussian Splatting“
>
> [3] Cen et al. (2024). "Segment Anything in 3D with Radiance Fields"
>
> [4] Wang et al. (2021). “A Comprehensive Survey and Experimental Comparison of Graph-Based Approximate Nearest Neighbor Search“

---

> > ### Author Response · Authors · 2024-11-22
> > **Answer to Reviewer N4ZX (2/2)**
> >
> > > *Is it possible to draw some theoretical connection of graph diffusion to spectral clustering methods, and therefore give some insight of the converging point of $g_t$?*
> >
> > There is indeed a connection to spectral clustering, as the iterates $g_t$ increasingly align with the span of the dominant eigenvectors of the similarity matrix. However, full convergence is not necessarily desirable in two cases that can happen in practice: 1) for a single connected component, the dominant eigenvector is constant and uninformative; 2) with many isolated components, there are too many dominant eigenvectors to interpret. To address this, we stop iterating before convergence (a soft approximation) and initialize $g_0$ to lie in the subspace corresponding to the target object. Note that for the segmentation task, regularizing with anchor similarities makes the process robust to the number of iterations.
> >
> > > *In figure 2, the rolling stair are separated from the ceiling in the multi-view case while merged to the latter in the single-view case, could the authors provide some explanation on such behavior?*
> >
> > The single-view visualization is a bilinear upsampling of DINOv2’s patch embeddings. At the patch level, representations of the ceiling and stairs are mixed, resulting in a globally green color. Aggregating representations from multiple views allows unmixing the semantic information contained in each patch embedding and attributing the right semantics to each pixel.
> >
> > > Using feature super-resolution (e.g., Feature-up in ICLR 2024 by Fu et al.)
> >
> > This is a very relevant suggestion. We tried introducing FeatUp in our pipeline, however the code available is only supported on ViT-S, which has much weaker generalization capabilities than ViT-g. The quality of ViT-S features combined with FeatUp lags behind that of ViT-g alone. As an example, we obtain an IoU of 81.7 for single-view segmentation on SPIn-NeRF using DINOv2's ViT-S combined with FeatUp, which is below our IoU of 88.3 obtained using DINOv2's ViT-g.

---

> > > ### Comment · Reviewer_N4ZX · 2024-11-28
> > > **On connection to spectral clustering.**
> > >
> > > 1) I am not sure if the dominant eigenvector would be constant, could the author please shed some light on this argument?
> > > 2) Arguing full convergence is less desirable is strange to me, early stoping strategy would evidently make the method less usable. A clear, feasible solution obtained in finite steps has long been desired in any algorithm design. I would recommend to rephrase and formulate the *true* convergence conditions more clearly.

---

> > > > ### Author Response · Authors · 2024-11-29
> > > >
> > > > > I am not sure if the dominant eigenvector would be constant. Could the author please shed some light on this argument?
> > > >
> > > > The dominant eigenvector(s) of the row-normalized matrix $\tilde{A} = D^{-1}A$ are constant on connected components of the graph; see Proposition 2 in [1], for example.
> > > >
> > > > This can be seen through the graph Laplacian $L = D - A$ and its associated quadratic form:
> > > > $
> > > > x^\top L x = \frac{1}{2} \sum_{(i, j)\, \text{neighbors}} A_{ij} (x[i] - x[j])^2.
> > > > $
> > > > This formulation demonstrates both that the eigenvalues of $L$ are non-negative and that any vector constant across the connected components is an eigenvector for $L$ with eigenvalue 0 (and hence an eigenvector for $\tilde{A}$ with eigenvalue 1).
> > > >
> > > > Thus, if the graph is connected, there is one dominant eigenvector—the vector of ones. If there are $N$ connected components, the dominant subspace is of dimension $N$ (allowing one independent value per connected component).
> > > >
> > > > ---
> > > > > Arguing that full convergence is less desirable seems strange to me. An early stopping strategy would evidently make the method less usable. A clear, feasible solution obtained in finite steps has long been desired in algorithm design. I recommend rephrasing and formulating the true convergence conditions more clearly.
> > > >
> > > > Indeed, you are correct, and we apologize for the confusion. Full convergence *is* desirable if the target object corresponds to a cleanly separated connected component in the graph $A$. In this scenario, the dominant eigenvector to which the algorithm converges provides an accurate segmentation mask.
> > > >
> > > > In cases where there is leakage in the graph, the diffusion process remains robust to the number of iterations **if the flow within the object is significantly larger than the flow at the object's boundaries**. This property ensures that the object corresponds to a *low-conductivity set* as defined in [2].
> > > > We enhance this robustness by "regularizing" the graph edges with anchor similarities ($P$ in the paper), which effectively cuts the flow to the background. We have empirically found that the **results remain almost identical with iterations ranging from 50 to 1000**.
> > > >
> > > > As for the mathematical convergence conditions, the power iteration method converges exponentially fast to the top eigenvector as long as the initial vector $g_0$ is not orthogonal to it (see for instance Chapter 7, Theorem 7.5 in [3]).
> > > > Given that $g_0$ is an initial semantic guess for the 3D segmentation mask, it already aligns well with the eigenvector.
> > > >
> > > > We hope these clarifications address your concerns, and we are happy to discuss this further.
> > > >
> > > > ---
> > > >
> > > > ### References
> > > > [1] Chunpai Wang, *Spectral clustering*, 2016
> > > >
> > > > [2] Meila and Shi, *Learning Segmentation by Random Walks*, NIPS 2000
> > > >
> > > > [3] Peter Arbenz, *Numerical Methods for Solving Large Scale Eigenvalue Problems*, Lecture notes (https://people.inf.ethz.ch/arbenz/ewp/Lnotes/chapter7.pdf)

---

### Official Review · Reviewer_sBLh · 2024-11-05

**Soundness:** 2
**Presentation:** 1
**Contribution:** 2
**Rating:** 3
**Confidence:** 3

**Summary:**

This paper is primarily concerned with generating 3D feature representations of a scene given multiple images of a stationary scene captured from different viewpoints. The main idea proposed here is that of extracting visual features from the multiple images (using a pretrained general purpose foundational model) and then fusing them with the scene representation computed using the 3D Gaussian splatting optimizer, i.e. fully calibrated camera poses and a set of 3D Gaussians with all the relevant parameters (position, covariance, size, opacity, spherical harmonics coefficients).

Specifically, feature vectors are computed and associated with the 3D centroids for each Gaussian in the 3DGS scene representation. The authors propose to compute these features by an aggregation scheme that is implemented as weighted averaging of the feature vectors extracted in multiple images. First 2D features are computed using existing visual feature extraction models. Features extracted for patches corresponding to the same Gaussian in multiple views are combined using a weighted averaging scheme presented in the paper. The 3D features associated with the 3DGS representation are then used in downstream tasks such as 3D segmentation (or multi-view segmentation) for which certain existing graph diffusion techniques are leveraged.

The experimental results focus on segmentation performance on the Spin-NeRF and NVOS datasets where the method achieves comparable performance to existing techniques.

**Strengths:**

The proposed approach to lift 2D features to 3D is efficient and avoids expensive iterative optimization schemes. The approach appears to be simple to implement and could be effective for downstream tasks such as 3D segmentation of stationary scenes where multiple views of the scene are available.

**Weaknesses:**

My main concerns about this work revolves around:

(1) Low novelty:
- It was unclear to me to what extent the main idea of lifting features from images to 3DGS point couds was novel compared to existing approaches in the literature that have explored scene editing given a 3DGS reconstruction of a scene. The weighted averaging and aggregation scheme described here appears to be very similar to what was proposed in prior work such as Chen et al. 2024.

- The paper mostly focuses on the 3D segmentation application and shows the their proposed ideas work well for that application. They experiment with different features (DINOv2 vs SAM, etc.). However, the authors are not the first to segment or edit 3D Gaussian scenes. This was also explored in the prior work of Chen et al 2024 although they focused on binary segmentation, unlike in this work.

(2) More thorough comparison with existing approaches needed:
The main claimed contribution in the work is towards the technical approach for computing 3D features from 2D features using an efficient approach (referred to as uplifting in the paper). There are other approaches in the literature that have been explored to accomplish the same step. Those optimization based methods are cited and discussed [Kerr et al 2023, Zuo et al 2024] but there is no quantitative comparison between the proposed method and those related works. This makes it difficult to assess the contribution and impact of the novel ideas that are presented here.

(3) Overall presentation has room for improvement:
The presentation and technical exposition could be improved for better clarity and presentation. I think the core objectives and the problem being tackled in the paper could have been explained better in the abstract and introduction. In the current version of the manuscript, it is difficult to understand what the authors precisely mean by “uplifting visual features to 3D scenes represented by Gaussian Splatting”. The abstract claims that “a simple yet effective aggregation technique yields excellent results.” – but the author have not explained what is being aggregated and for what purpose and what the success criteria is. The authors then talk about competitive segmentation results without clearly explaining the connection between the segmentation task and the 3D reconstruction or 3D scene representation task. Reading Section 1 (Introduction) doesn't give a concrete idea of the main objective, the key motivations and where the novelty lies.

**Questions:**

Is the graph diffusion framework based on existing work? If so, the appropriate work in the literature should be cited for that. Otherwise, it would be best to clarify that the formulation is novel and developed by the authors themselves for the 3d segmentation task.

"Let I1, . . . , Im be a set of 2D frames from a 3D scene and d1, . . . , dm the corresponding viewing directions." -- the authors should clarify what they mean by viewing directions, and whether they are referring to camera poses, here? The phrase viewing direction is used throughout the paper and makes it difficult to understand the underlying mathematical operations that are being performed.

The mathematical notation is difficult to follow. The expression \hat{F} = (\hat{F}_d, [) (lines 188) are not well explained. Similarly, the steps to construction the W and D matrices also need to clearly described.

---

> ### Author Response · Authors · 2024-11-22
> **Answer to Reviewer sBLh (1/2)**
>
> We would like to thank the reviewer for their detailed feedback on the positioning of our work with respect to the literature. Below, we address their remarks, including the question of the novelty of our proposed uplifting method and graph diffusion process, and present new experimental results on open-vocabulary object detection to compare with the prior works mentioned by the reviewer.
>
> > Novelty compared to Chen et al. 2024 [1]
>
> As mentioned in the method section, there is some similarity between Chen et al 2024 [1] and our uplifting strategy. Yet, our work is quite different. Chen et al. focus on scene editing, relying on rough binary segmentation masks (high recall) to capture objects of interest (see Figure 2 in their paper). While their approach is effective for their goal, it struggles with precise segmentation (high precision) across multiple categories. We evaluated Chen et al’s aggregation on the segmentation tasks, and obtained  87.3 / 88.6 IoU on NVOS and 78.5 / 92.5 on SPIn-NeRF with DINOv2 / SAM2 respectively, which is far behind our reported results. We observed that results on SPIn-NeRF with DINOv2 are particularly poor for 360-degree scenes and scenes with a high specularity modeled by floaters in the Gaussian Splatting reconstruction. We will include these additional results as well as qualitative comparisons of projected 3D SAM and DINOv2 features in our revised version.
>
> Even though the difference may seem minor at first, performing the increasingly common 3D-distillation of 2D foundation models with a simple and generic summation – requiring no optimization – is likely to change standard practice in the field.
>
> [1] Chen et al. (2024). “GaussianEditor: Swift and Controllable 3D Editing with Gaussian Splatting”
>
>
> > Quantitative comparison with methods Kerr et al 2023, Zuo et al 2024.
>
> The works from Kerr et al. (LERF) [1] and Zuo et al. (FMGS) [2] jointly uplift CLIP and DINO features for open-vocabulary localization tasks. Our work focused on uplifting DINOv2 and SAM vision models, but is indeed generic enough to extend to CLIP. To provide the requested comparison, we uplifted CLIP feature maps and evaluated their quality on the LERF localization task. Inspired by LERF [1] and FMGS [2] who use DINOv2 as a regularization, we uplift DINOv2 features jointly with CLIP and refine CLIP features through graph diffusion with a transition matrix defined by DINOv2 pairwise feature similarities.
>
> Below we report our results with and without graph diffusion as well as estimated total running times. Our method yields results comparable with the litterature while being significantly faster.
>
>
> | Test Scene | LERF [1]	 | LangSplat [3]	| FMGS [2]	 | Ours (no diffusion | Ours (diffusion)
> |------------------|---------------------|------------------|-------------------|------|---------
> | ramen    	|     62.5  	|    73.2 	 |       90.0 	 |  	71.8	  |   	77.5	|
> | figurines 	|    79.5	 |   80.4  	|         89.7   	 |	42.9	 |       	78.6	|
> | teatime   	|    93.8	 |   88.1	|         93.8   	 |       	89.8	 |	94.9	|
>  | waldo_kitchen |    81.5 	|   95.5 	|      92.6          |    	81.8    	 |   	86.4	|
>  | overall	 |    80.3 	|   84.3	 	|       91.5     	| 	     71.6   	 |	84.4	|
>  | average time (minutes) |    	 45  |    105 	|      100        	|  	7  	| 	9    |
>
>
> *References*
>
> [1] Kerr et al. (2023). “LERF: Language-Embedded Radiance Fields”
>
> [2] Zuo et al. (2024). “FMGS: Foundation Model Embedded 3D Gaussian Splatting for Holistic 3D Scene Understanding”
>
> [3] Qin et al. (2024). “LangSplat: 3D Language Gaussian Splatting“

---

> > ### Author Response · Authors · 2024-11-22
> > **Answer to Reviewer sBLh (2/2)**
> >
> > > *The authors are not the first to segment or edit 3D Gaussian scenes.*
> >
> > Indeed, others have already tackled segmentation of 3D Gaussian scenes, some of them mentioned in the second paragraph of the related work section.
> > The focus of the paper was i) to show that a simple aggregation works on par with more complex approaches employed in works to which we compare in Tables 1 and 2, and ii) that DINOv2 yields segmentation results as good as those obtained with SAM which, to the best of our knowledge, we are the first to demonstrate. We will clarify these points.
> >
> > > *Is the graph diffusion framework based on existing work?*
> >
> > The graph diffusion framework for segmentation is grounded in basic graph theory and is conceptually related to spectral clustering [2] and graph cut algorithms such as NCut [1]. Previous works [3, 4, 5] have explored using patch similarity graphs for 2D segmentation in an unsupervised manner. However, to the best of our knowledge, our formulation is novel in the context of 3D object segmentation. We will clarify this point.
> > Key novelties in our approach include:
> > 1. **Prompt-driven diffusion**: We diffuse a user-defined prompt (e.g., a point, scribble, or reference mask), enabling segmentation of any object in the scene. This contrasts with [3, 4, 5], which focus only on foreground/background segmentation.
> > 2. **Anchor-point similarities**: We incorporate anchor-point similarities into the edge weights to constrain diffusion and prevent it from spreading into the background.
> > 3. **Efficient computation**: Given the large size of the similarity matrix in 3D, we approximate spectral embedding using the power method rather than directly computing eigenvectors.
> >
> > *References*:
> >
> > [1] Shi and Malik (2000). “Normalized Cut and Image Segmentation.”
> >
> > [2] Ng et al. (2002). "On spectral clustering: analysis and an algorithm.”
> >
> > [3] Simeoni et al. (2021). “Localizing Objects with Self-Supervised Transformers and no Labels.”
> >
> > [4] Wang et al. (2022). “Self-Supervised Transformers for Unsupervised Object Discovery Using Normalized Cut.”
> >
> > [5] Melas-Kyriazi et al. (2022). “Deep Spectral Methods: A Surprisingly Strong Baseline for Unsupervised Semantic Segmentation and Localization.”
> >
> > > *Reading the abstract and introduction doesn't give a concrete idea of the main objective, the key motivations and where the novelty lies.*
> >
> > Please refer to the proposed abstract in the general response. We will include it in the revised version as well an updated introduction that better contextualizes our work.
> >
> > > *The authors should clarify what they mean by viewing directions and whether they are referring to camera poses here.*
> >
> > We are indeed referring to camera poses, and will fix the formulation in the paper.
> >
> > > Notation
> >
> > * We will clarify the fact that the expression $\hat{F}$ in l.188 is a 2D matrix containing all (flattened) 2D feature maps generated for all cameras poses, with $\hat{F}_{d,p}$ pointing to the feature of pixel $p$ viewed from camera pose $d$.
> > *  We will clarify  that $W$ and $D$ are not explicitly constructed. Instead, they are computed by calling the forward rendering function for Gaussian Splatting and replacing the color vectors by the feature vectors. All these operations are performed within the CUDA rendering process.

---

> > > ### Comment · Reviewer_sBLh · 2024-11-26
> > >
> > > Thanks for the clarification. Is the current formulation not a simple application of spectral clustering, then? Would it be possible to revise the paper accordingly. What is the benefit of referring to this part as a graph diffusion formulation. The term diffusion is more generic but I don't see what advantage there is to use that term instead of spectral clustering?
> > >
> > > Power methods for speeding up eigen decomposition is well-known. Then I don't see how efficient computation can be claimed as a key novelty (in the graph diffusion stage of the proposed approach).

---

> > ### Comment · Reviewer_sBLh · 2024-11-26
> >
> > Thanks for clarifying the differences with Chen et al. 2024 [1]. In my opinion these differences are indeed quite minor. Empirically the new aggregation method performs better. So better empirical performance is a plus in terms of contributions but technical insights into what leads to the better empirical results is lacking, so I still consider this as a case of low/incremental novelty.

---

> ### Author Response · Authors · 2024-11-28
>
> > Thanks for clarifying the differences with Chen et al. 2024 [1]. In my opinion these differences are indeed quite minor. Empirically the new aggregation method performs better. So better empirical performance is a plus in terms of contributions but technical insights into what leads to the better empirical results is lacking, so I still consider this as a case of low/incremental novelty.
>
>
> The key technical insight lies in the compatibility between optimization-based approaches and our proposed rule, which can be interpreted as a gradient descent step preconditioned by the square root of the Hessian's diagonal. This makes it an effective minimization step for the reconstruction loss, in contrast to the method described by Chen et al. (2024). This distinction has a significant impact. Do you see any points of disagreement with the arguments we have presented above?

---

> ### Author Response · Authors · 2024-11-28
>
> > Thanks for the clarification. Is the current formulation not a simple application of spectral clustering, then? Would it be possible to revise the paper accordingly. What is the benefit of referring to this part as a graph diffusion formulation. The term diffusion is more generic but I don't see what advantage there is to use that term instead of spectral clustering?
> Power methods for speeding up eigen decomposition is well-known. Then I don't see how efficient computation can be claimed as a key novelty (in the graph diffusion stage of the proposed approach).
>
> Diffusion on graphs is a terminology that has been around for about 20 years
> [1,2], which refers to a regularization mechanism. As discussed above, this
> mechanism is rooted in spectral graph theory and used in spectral
> clustering/normalized cuts/Laplacien eigenmaps techniques [3,4,5].
>
> In our work, we leverage this regularization mechanism (diffusion), but without
> performing any clustering steps or computing explicit spectral embeddings. This
> supports the use of the term ``diffusion''. The revised version of our paper includes a discussion on the connections to spectral theory.
>
> Regarding the novelty here, we are the first to show that such a mechanism is helpful for feature uplifting in 3D scenes, enriching masks/features with 3D geometrical information represented by the graph topology.
>
> Does this explanation address your concern?
>
> > Power methods for speeding up eigen decomposition is well-known. Then I don't see how efficient computation can be claimed as a key novelty (in the graph diffusion stage of the proposed approach).
>
> Our only claim here is that we propose an effective method for uplifting features, which is highly efficient—an order of magnitude faster than FMGS or LansSplat (as demonstrated in the table above)—thanks to the use of a simple uplifting rule and a fast optimization algorithm for graph diffusion (the power method). We would greatly appreciate knowing if you agree with this claim.
>
> *[1] Kondor and Lafferty. Diffusion kernels on graphs and other discrete structures. ICML, 2002.*
>
> *[2] Smola and Kondor. Kernels and Regularization on Graphs. Learning theory and kernel machines. 2003.*
>
> *[3] Belkin and Niyogi. Laplacian Eigenmaps and Spectral Techniques for Embedding and Clustering. NIPS, 2001.*
>
> *[4] Shi and Malik. Normalized cuts and image segmentation. PAMI, 2000.*
>
> *[5] Melia and Shi. Learning Segmentation by Random Walks. NIPS. 2000.*

---

### Author Response · Authors · 2024-11-22
**Joint answer to the reviewers**

### Joint answer

First, we would like to thank all reviewers for their careful reading and insightful comments. We have provided individual answers to each reviewer below. Those contain our current results on the ongoing experiments we are running to respond to the reviewers' questions and requests. The final results will be reported in the revised version of our paper that we plan to upload towards the end of the discussion period.
The additional experiments address comments from reviewers *sBLh* and *jxL3* on extending the experimental evaluation and running time comparaisons.
We extended our evaluation to the task of open-vocabulary object detection. To this end, we uplifted CLIP features and refined them using graph diffusion with DINOv2 features.
Our approach achieves results comparable to the litterature while drastically reducing run times.

In the revised version of our paper we will:
- include these additional experiments on the task of open-vocabulary object detection, showing that our method does not only work for DINOv2 and SAM but also for CLIP (*answering sBLh and jxL3*)
- include comparative reports on running times (*answering jxL3 and N4ZX*)
- add qualitative comparisons with other baselines as well as visualizations of scene editing using 3D segmentation (*answering jxL3*)
- discuss the theoretical inspiration of our graph diffusion process (*answering QRFz and N4ZX*)
- add a discussion on the scope and limitation of the method as concluding remarks based on the anwsers we provide to N4ZX and QRFz.
- clarify notations, add an illustration of the feature generation process (*answering sBLh and jxL3*), and revise the introduction and abstract so as to more clearly state the context, objectives and key contributions (*answering sBLh*). The new version of the abstract is available below.


### Abstract
The advent of powerful vision foundation models such as DINO, SAM, and CLIP has spurred interest in extending their capabilities to 3D tasks. These off-the-shelf models produce rich 2D features for tasks such as text embedding, semantic understanding, and segmentation at low computational cost.  Motivated by a recent breakthrough in 3D scene modeling, Gaussian Splatting, which enables real-time, high-resolution rendering of 3D features, we suggest a novel method to distill — or *uplift* — 2D image features into 3D.
While common approaches rely on minimizing a reconstruction loss, we instead propose a simple, efficient and parameter-free feature aggregation method that bypasses the need for costly optimization.  This method achieves comparable performance on multiple downstream tasks while offering significant speed-ups.

 When applied to semantic masks generated by the Segment Anything Model (SAM), our approach achieves segmentation quality comparable to state-of-the-art techniques. Remarkably, we also achieve competitive segmentation results using generic features from DINOv2, despite DINOv2 not being trained on millions of annotated segmentation masks like SAM. When applied to vision-language features from CLIP, our method achieves competitive results on open-vocabulary, language-based object detection. As a key contribution, we leverage the 3D scene geometry and the rich  similarities induced by DINO embeddings with a graph diffusion process, which refines the localization of coarse semantic features such as CLIP features or segmentation masks.

---

### Comment · Area_Chair_Samn · 2024-11-25
**Please read the rebuttal and response**

Dear Reviewers,

Thanks again for serving for ICLR, the discussion period between authors and reviewers is approaching (November 27 at 11:59pm AoE), please read the rebuttal and ask questions if you have any. Your timely response is important and highly appreciated.

Thanks,
AC

---

### Author Response · Authors · 2024-11-28
**Revised manuscript**

We thank again all the reviewers for their feedback. We have uploaded a revised version of our submission (with changes in blue) that includes:
- our additional experiments on the task of **open-vocabulary object detection** in Sections 4.2 and 5.4 (answering *sBLh* and *jxL3*)
- comparative reports on **running times** in Sections 5.4 and B.2 (answering *jxL3* and *N4ZX*)
- **qualitative comparisons** with other baselines in Sections C.2 and C.3 as well as visualizations of scene editing using 3D segmentation in Figure 7 (answering *jxL3*)
- a discussion on the theoretical inspiration of our graph diffusion process in Section 3.3 (answering *QRFz* and *N4ZX*)
- a discussion on the scope and limitation of the method in Section 6 (answering *N4ZX* and *QRFz*)
- clarified notations in Section 3, an illustration of the feature generation process in Figure 4 (answering *sBLh* and *jxL3*)
- a revised introduction and abstract (answering *sBLh*).

---

### Author Response · Authors · 2024-12-02
**Consolidated results on LERF segmentation task**

We thank the reviewers once again for engaging in the discussion during this rebuttal process. We have extended the experimental evaluation on open-vocabulary object localization by including additional results for the LERF Segmentation task [1], evaluated on the more challenging version introduced by LangSplat [2]. This task consists in predicting segmentation masks for different textual queries and complements our previous results on LERF Localization.

Our segmentation masks are predicted using SAM with point prompts extracted from CLIP relevancy maps. This aligns with FMGS's [3] approach (though FMGS does not evaluate this task) and adds only a small computational overhead during evaluation (1 to 4 seconds per test image). On this task, we outperform LangSplat by a significant margin.

These additional experimental results further confirm the effectiveness of our approach, reinforcing that its simplicity does not come at the cost of accuracy. We hope this addresses Rev. *QRFz*'s concerns and convinces Rev. *jxL3* and *sBLh* that our contribution has the potential to significantly impact the community.

| LERF Segmentation (IoU)        | **LERF**  | **LangSplat**  | **Ours**             |
|--------------------|----------------|----------------|-----------------------|
| ramen              | 28.2           | **51.2**           |    50.2                   |
| figurines          | 38.6           | 44.7           |     **55.1**                  |
| teatime            | 45.0           | 65.1           |   **69.2**                    |
| waldo_kitchen      | 37.9           | 44.5           |  **59.0**                     |
| **overall**        | 37.4       | 51.4       |  **58.4**                     |
| **average time (minutes)** | 45 | 105       | **9**                |
-------------------------------------------------------------------

*[1] Kerr et al. (2023). “LERF: Language-Embedded Radiance Fields”*

*[2] Qin et al. (2024). “LangSplat: 3D Language Gaussian Splatting“*

*[3] Zuo et al. (2024). “FMGS: Foundation Model Embedded 3D Gaussian Splatting for Holistic 3D Scene Understanding”*

---

### Meta-Review · Area_Chair_Samn · 2024-12-17

**Metareview:**

This paper proposes a method that lift 2D features from foundational models (e.g., CLIP, SAM) to learned 3D Gaussians for a scene. The main idea is to apply weight aggregation on the features of each pixel, as well as a diffusion mechanism that diffuse features to nearby Gaussians. The propose approach is efficient compared to optimization-based methods. The weaknesses of the paper include low novelty (Reviewer sBLh), confusing presentation (Reviewer sBLh, QRFz), effectiveness and robustness of the method (Reviewer N4ZX, jxL3). The AC agrees with the reviewers that the paper has limited novelty and needs improvements on better differentiation with other methods, presentation and evidence to show the robustness of the proposed approach, thus the paper is recommended for rejection. The authors are encouraged to improve the paper given the comments of the reviewers.

**Additional Comments On Reviewer Discussion:**

During rebuttal, the reviewers raised questions about
- the similarity of the paper with existing works (Reviewer sBLh)
- presentation issues (Reviewer sBLh, QRFz)
- effectiveness and robustness of the method (Reviewer N4ZX, jxL3)
- questions about technical details

Authors have responded to all of the questions. After rebuttal, the paper receives two rejects, one borderline reject and one borderline accept. Given that not all issues raised by the reviewers are fully addressed, the paper is recommended for rejection for this time.

---

### Decision · Program_Chairs · 2025-01-22

Reject